# Notch3-dependent β-catenin signaling mediates EGFR TKI drug persistence in EGFR mutant NSCLC

Rajeswara Rao Arasada[1], Konstantin Shilo[2], Tadaaki Yamada[3], Jianying Zhang[4], Seiji Yano[3], Rashelle Ghanem[1], Walter Wang [1], Shinji Takeuchi[3], Koji Fukuda[3], Nobuyuki Katakami[5], Keisuke Tomii[6], Fumitaka Ogushi[7], Yasuhiko Nishioka[8], Tiffany Talabere[1], Shrilekha Misra[9], Wenrui Duan[1], Paolo Fadda[1], Mohammad A. Rahman[10], Patrick Nana-Sinkam[10], Jason Evans[1], Joseph Amann[1], Elena E. Tchekneva[1], Mikhail M. Dikov[1] & David P. Carbone[1]

EGFR tyrosine kinase inhibitors cause dramatic responses in EGFR-mutant lung cancer, but resistance universally develops. The involvement of β-catenin in EGFR TKI resistance has been previously reported, however, the precise mechanism by which β-catenin activation contributes to EGFR TKI resistance is not clear. Here, we show that EGFR inhibition results in the activation of β-catenin signaling in a Notch3-dependent manner, which facilitates the survival of a subset of cells that we call "adaptive persisters". We previously reported that EGFR-TKI treatment rapidly activates Notch3, and here we describe the physical association of Notch3 with β-catenin, leading to increased stability and activation of β-catenin. We demonstrate that the combination of EGFR-TKI and a β-catenin inhibitor inhibits the development of these adaptive persisters, decreases tumor burden, improves recurrence free survival, and overall survival in xenograft models. These results supports combined EGFR-TKI and β-catenin inhibition in patients with EGFR mutant lung cancer.

[1] Department of Internal Medicine, Division of Medical Oncology, The Ohio State University Medical Center, Columbus, OH 43210, USA. [2] Department of Pathology, The Ohio State University Medical Center, Columbus, OH 43210, USA. [3] Division of Medical Oncology, Kanazawa University Cancer Research Institute, Kanazawa 920-0934, Japan. [4] Center for Biostatistics, The Ohio State University Medical Center, Columbus, OH 43210, USA. [5] Division of Integrated Oncology, Institute of Biomedical Research and Innovation, Kobe 650-0047, Japan. [6] Department of Respiratory Medicine, Kobe City Medical Center General Hospital, Kobe 650-0047, Japan. [7] Division of Pulmonary Medicine, National Hospital Organization National Kochi Hospital, Kochi 780-8077, Japan. [8] Department of Respiratory Medicine and Rheumatology, Graduate School of Biomedical Sciences, Tokushima University, Tokushima 770-8503, Japan. [9] Department of Internal Medicine, The Ohio State University Medical Center, Columbus, OH 43210, USA. [10] Division of Pulmonary, Allergy, Critical Care and Sleep Medicine and the Center for Critical Care Medicine, The Ohio State University Medical Center, Columbus, OH 43210, USA. Correspondence and requests for materials should be addressed to R.R.A. (email: Rajeswara.arasada@osumc.edu) or to D.P.C. (email: david.carbone@osumc.edu)

Non-small cell lung cancer (NSCLC) accounts for 85% of all lung cancer incidence and is the leading cause of cancer death[1]. In the US ~15% of the patients with NSCLC have tumors associated with "driver" mutations in the EGFR gene that demonstrate major clinical responses to EGFR tyrosine kinase inhibitors (EGFR TKIs)[2]. However, EGFR TKI therapy results in responses of variable depth and duration and is not curative because complete tumor eradication is never achieved. Some of this variability is due to pre-existing EGFR T790M mutations that are resistant to first generation TKIs, but even with newer generation drugs that are highly effective against this subclone (such as osimertinib), a subpopulation of cells survives, enabling the eventual development of other resistance mechanisms[3–7]. How this subpopulation of EGFR mutant lung cancer cells avoids eradication after complete inhibition of EGFR is unclear[8].

We and others have reported that erlotinib treatment rapidly enriches residual tumors for a drug persistent population[9,10]. We have shown that this process is sensitive to inhibition of Notch3 and identified a novel physical association between the EGFR receptor and the Notch3 protein that is indispensable for the induction of drug persistent cells (DPCs), which have many properties of stem-like or progenitor cells[9]. Based on our data and those of others, Notch3 (but not the other Notch receptors) has a pivotal role in the maintenance of a progenitor population in human lung cancer cells and also in KRAS driven mouse lung tumors[9,11,12]. However, the precise mechanism by which Notch3 maintains this progenitor phenotype is not understood, and specific targeting of this pathway has been a challenge.

Activation of canonical Notch signaling requires interaction with a ligand on a signal-sending cell, exposure of specific protease sites, and cleavage of the receptor to release the Notch intracellular domain (NICD). The NICD translocates into the nucleus and interacts with the CSL transcription factor complex to activate Notch target genes, such as the Hes-family and Hey-family members[13]. Non-canonical signaling is more complex and less well studied. One of the non-canonical activities of the Notch1 receptor is its effect on β-catenin activity. Notch1 activation has been shown to inhibit Wnt/β-catenin signaling through physical association with β-catenin in both mouse and stem cell models[14]. Notch3 has been shown to regulate Wnt signaling in mammary cell differentiation by controlling Frizzled receptor expression in a CSL-independent manner[15,16]. In T-cell leukemia, Notch3 was shown to activate NF-kB through its association with the pre-T cell receptor (pre-TCR) pTα chain[15,16].

Altered Wnt/β-catenin signaling has been reported to play a pro-tumorigenic role in many cancers. Up to 80% of colon cancer tumors have loss of function mutations in APC, which leads to activation of β-catenin and increased tumorigenesis. In NSCLC, APC mutations are rare. However, mutations in β-catenin have been recently reported in patients that are resistant to EGFR TKI therapy and in EGFR mutant metastatic lung cancers[17,18]. Altered Wnt/β-catenin pathway-related genes have also been reported and are associated with poor prognosis[19]. Canonical Wnt signaling has been demonstrated to play a role in the survival of EGFR mutant NSCLC during EGFR TKI treatment and more recently, studies have also showed that β-catenin plays a role in drug resistance associated with secondary mutations in the EGFR gene[20,21]. This highlights a critical role for β-catenin in the upregulation of survival pathways with EGFR TKI therapy[20–22]. Nonetheless, the role of β-catenin in the early acquisition of adaptive persistence after treatment with EGFR TKIs has not been described. Moreover, the role of β-catenin activation in mediating the observed variability in the depth and duration of initial response is unknown.

In order to improve the outcomes of patients with mutant EGFR NSCLC, we need to define and target the basis of this variable initial response and the mechanisms by which tumor cells persist through the initial phase of therapy. Our in vitro model system of erlotinib-induced DPCs has specifically defined Notch3 as a critical mediator of this effect, but there are no available agents to specifically target the non-canonical activity of Notch3, so we sought to identify potentially targetable pathways that are controlled by Notch3 in this process. In doing so, we identify a novel signaling pathway involving Notch3 and β-catenin that is associated with EGFR resistance, and define this pathway as a key targetable mediator of the rapid development of Notch3-dependent EGFR TKI adaptive persistence.

## Results

**EGFR TKI induces targets of β-catenin in a Notch3-dependent manner.** We have previously shown that the early induction of DPCs that have stem-like properties immediately after treatment with erlotinib requires Notch3, but not Notch1[9]. In fact, Notch1 knockdown increases the fraction of these cells (Supplementary Fig. 1a). We took advantage of N3/N1 specificity to help define downstream pathways responsible for this effect by looking for transcripts that were differentially expressed in EGFR mutant HCC4006 cells 6 days after addition of erlotinib with and without Notch3 knockdown, but were not differential (or differential in the opposite direction) between erlotinib-treated cells and those with Notch1 knockdown. This would specifically define pathways dependent on Notch3 and not Notch1. We analyzed cells transfected with non-targeting control siRNA (NTC), Notch1 siRNA (N1) or Notch3 siRNA (N3) and treated with DMSO (vehicle) or erlotinib. Western blot analysis of Notch1 and Notch3 showed that knockdowns were specific for the respective proteins and appeared to be complete (Supplementary Fig. 1b). We applied multi-level stringent selection criteria to identify DPC-specific targets by initially identifying those genes that are upregulated with erlotinib treatment. Using our knowledge that Notch1 knockdown increases the stem cell phenotype and Notch3 knockdown eliminates the stem cell phenotype, additional selection was performed by identifying those genes whose expression is further enhanced with Notch1 knockdown but decreased with Notch3 knockdown and considered them as DPC-specific genes. Interestingly, the analysis did not identify transcriptional targets of Notch, such as Hes or Hey family members (Supplementary Tables 1 and 2). Instead, we identified PAI1 and MMP7, both transcriptional targets of β-catenin, as the two top genes significantly upregulated upon erlotinib treatment and differentially altered with Notch1 or Notch3 silencing (Supplementary Tables 1 and 2)[23,24]. In addition, we found multiple other proteins, such as HIST1H, SAMD9L, ANKRD1, GADD45, HBEGF, ABI3BP, and CTGF that are regulated by β-catenin or are associated with β-catenin transcriptional activity and are reported to play a major role in the maintenance of stem cells (Supplementary Tables 1 and 2)[23–31].

We validated the changing expression levels of PAI1 and MMP7 using qRT-PCR. In agreement with the microarray data, these two targets were induced by erlotinib, exhibited enhanced expression with Notch1 knockdown, above what was seen with erlotinib alone, and decreased expression with Notch3 knockdown (Fig. 1a). These results confirm the observed Notch3-specific control of PAI1 and MMP7 expression with erlotinib treatment. This analysis shows that the rapid induction of DPCs after treatment with EGFR-TKIs is strongly associated with upregulation of β-catenin target genes or regulators, which are known to play a role in maintaining a stem-like phenotype, and that this effect is dependent on Notch3.

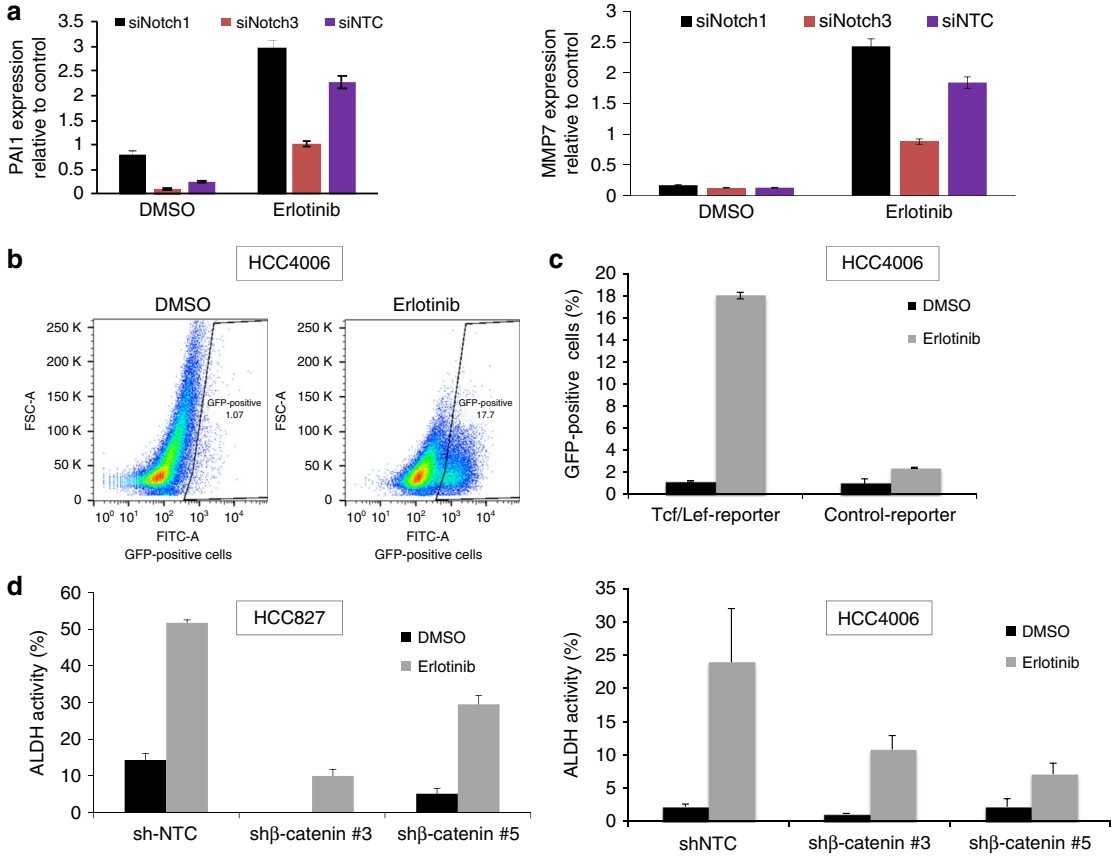

**Fig. 1** Identification and validation of regulators of EGFR TKI-mediated DPCs in EGFR mutant NSCLC cells. **a** qRT-PCR validation of PAI1 and MMP7 expression levels in HCC4006 cells. **b** HCC4006 cells expressing Tcf/Lef-GFP reporter were treated with DMSO or 0.1 μM erlotinib for 6 days and sorted by FACS for GFP fluorescence. **c** Histograms and quantification from triplicates was plotted and include a negative control with mutated Tcf/Lef consensus binding sites. The mutant reporter has no activity after erlotinib treatment. DMSO-treated cells were used as a base line control. **d** HCC827 (left) and HCC4006 (right) were stably infected with shRNAs targeting non-targeting control (shNTC) or β-catenin (shRNAs #3 and #5) were treated with DMSO or erlotinib and subjected to ALDH assay. For (**a**) and (**c**) error bars represent SD from biological triplicates. For (**d**) error bars represent SD from technical triplicates

**β-catenin is necessary for EGFR-TKI induction of ALDH activity**. To confirm a functional role of β-catenin in the induction of EGFR TKI persistence, we infected HCC4006 cells with a lentivirus containing wild type, (8xTOP) or mutant (8xFOP) consensus T-cell factor (TCF) DNA-binding sites that drive expression of GFP. HCC4006 cells with reporter were treated with DMSO or 0.1 μM of erlotinib for 6 days. EGFR TKI persistent cells were assayed for β-catenin transcriptional activity. Cells treated with DMSO served as a baseline to distinguish GFP high and low cell populations. We observed that cells treated with EGFR TKI showed a dramatic increase in transcriptional activity of HCC4006-TCF-GFP cells. Cells expressing mutant TCF-GFP did not show any activity as expected (Fig. 1b, c and Supplementary Fig. 2a, b).

We have previously shown that erlotinib treatment, while reducing the total number of cells, dramatically increases the fraction of ALDH-positive cells and increases pulmosphere formation in a Notch3-dependent manner. Our finding that erlotinib treatment rapidly increases β-catenin activity, suggested that β-catenin might mediate the induction of ALDH activity. To address this, HCC827 and HCC4006 cells infected with lentivirus expressing NTC or β-catenin shRNAs were treated with DMSO or 0.1 μM erlotinib for 6 days and subjected to ALDH assays. Loss of β-catenin expression significantly decreased the ALDH-

positive population (Fig. 1d and Supplementary Fig. 3a, b). These results implicate β-catenin as a critical mediator of EGFR-TKI-induced DPCs.

**Notch3 regulates β-catenin activity in a non-canonical manner**. To determine if the regulation of β-catenin activity by Notch3 is through canonical or non-canonical signaling, we overexpressed dominant negative mastermind (DN-MAML1), an inhibitor of canonical Notch activity, prior to induction of ALDH-positive cells with erlotinib treatment. The DN-MAML1 did not inhibit the EGFR-TKI induction of ALDH-positive cells demonstrating that Notch3 acts in a non-canonical fashion (Fig. 2a). As a control, we demonstrated that HCC827 cells expressing DN-MAML1 have reduced Notch transcriptional activity suggesting that the DN-MAML1 is active and capable of inhibiting canonical Notch transcriptional activity in this setting (Fig. 2b). We further performed analysis of HES1 expression in cells stimulated with control (basal) or Notch ligand, DLL1 (activated), which clearly showed that basal HES1 transcript levels are down with DN-MAML1, but not abolished strongly. On the other hand, DN-MAML1 was completely abolished following ligand-activated HES1 expression suggesting that DN-MAML1 is active in inhibiting the canonical Notch signaling (Fig. 2c).

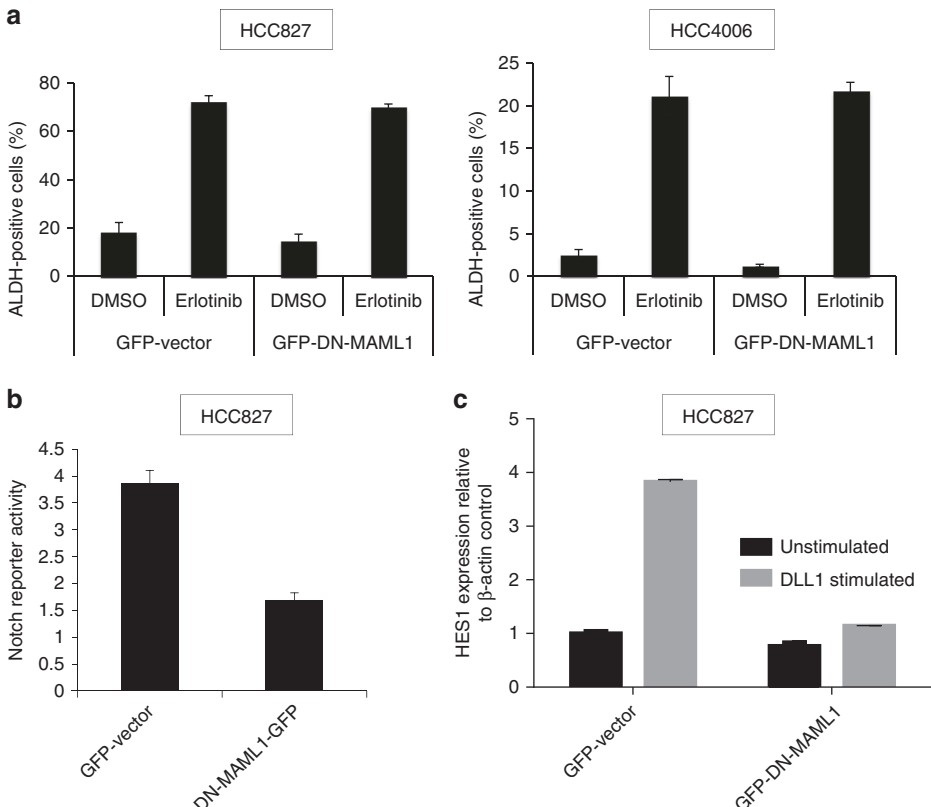

**Fig. 2** EGFR TKI-induced ALDH activity does not require MAML activity. **a** HCC827 (left) and HCC4006 (right) were stably infected with control vector or DN-MAML1 and treated with DMSO or erlotinib and subjected to ALDH assay. **b** HCC827 cells that were stably infected with control vector or DN-MAML1 were also treated stably infected with virus expressing notch operating promoter (NOP) reporter and NOP activity was assayed using FACS. **c** HCC827-Vector or HCC827-DN-MAML1 were grown on plates coated with DLL1-Fc chimera or control (PBS) for 48 h. Total RNA was isolated and HES1 expression was analyzed using qRT-PCR. For (**a**) and (**c**) error bars represent SD from biological triplicates. For (**a**) and (**b**) error bars represent SD from biological triplicates. For (**c**) error bars represent SD from technical triplicates

**EGFR-TKI induces an association between Notch3 and β-catenin**. Our observations suggesting that Notch3 regulates β-catenin in a non-canonical manner led us to hypothesize a physical association between Notch3 and β-catenin as the basis for this regulation. Co-immunoprecipitation analyses identified an association between Notch3 and β-catenin in HCC827 and HCC4006 cells (Fig. 3a, b). Furthermore, the degree of association between Notch3 and β-catenin was dramatically increased when cells were treated with erlotinib. To further understand whether the association is due to gamma secretase-dependent Notch3 cleavage, co-immunoprecipitation experiment was done in cells treated with EGFR TKI, gamma secretase inhibitor (GSI) alone or in combination. We found that EGFR TKI-mediated Notch3 association with β-catenin is sensitive to GSI suggesting that cleaved Notch3 (intracellular) could be associating with β-catenin (Fig. 3c). We then performed co-localization experiments to test the hypothesis that erlotinib treatment increases the association of Notch3 and β-catenin in the cytoplasm. In untreated cells β-catenin and Notch3 proteins were detected on the cell membrane and in the cytoplasm and weak co-localization of Notch3 and β-catenin was observed. However, when cells were treated with erlotinib, we detected a strong upregulation and co-localization of both Notch3 and β-catenin (Fig. 3d, e). Furthermore, we found that the co-localization was entirely cytoplasmic, providing further evidence that Notch3 plays a non-canonical role in regulating β-catenin transcriptional activity. In addition, EGFR-TKI-mediated up-regulation and co-localization of Notch3 and β-catenin was variable in the population, and highest only in a

fraction of cells strongly suggests that these cells are potentially the DPC fraction.

To determine the effect of EGFR-TKI treatment on the β-catenin protein levels and half-life, cells were treated with DMSO or erlotinib and analyzed using total and non-phospho (active) β-catenin antibodies. We identified an increase in activated form (non-phosphorylated) of β-catenin (Fig. 3a). We next determined the protein stability by cycloheximide treatment and found that when compared to control, EGFR-TKI treatment completely stabilized the non-phosphorylated, active β-catenin and no protein turnover was seen through the time course (Fig. 3b, c). To further understand the effect of Notch3 on β-catenin activation we performed cytoplasmic and nuclear fractionation and found that EGFR TKI treatment leads to increased nuclear β-catenin, both total and active forms, and decreased phospho β-catenin (inactive). Interestingly, we also found that EGFR TKI treatment leads to decreased nuclear Notch3 and increased cytoplasmic Notch3, which is in agreement with our hypothesis and supports our notion that cytoplasmic association between Notch3 and β-catenin is helping β-catenin stability. Overall, these findings demonstrate the non-canonical role for Notch3 where it activates β-catenin signaling rather than activating Notch signaling in a canonical manner, which would have required nuclear Notch3 in EGFR TKI-treated cells (Fig. 3d). These data suggest that one mechanism by which EGFR-TKI treatment induces β-catenin activity is that the association between Notch3 and β-catenin stabilizes the β-catenin protein, and thereby increasing its transcriptional activity.

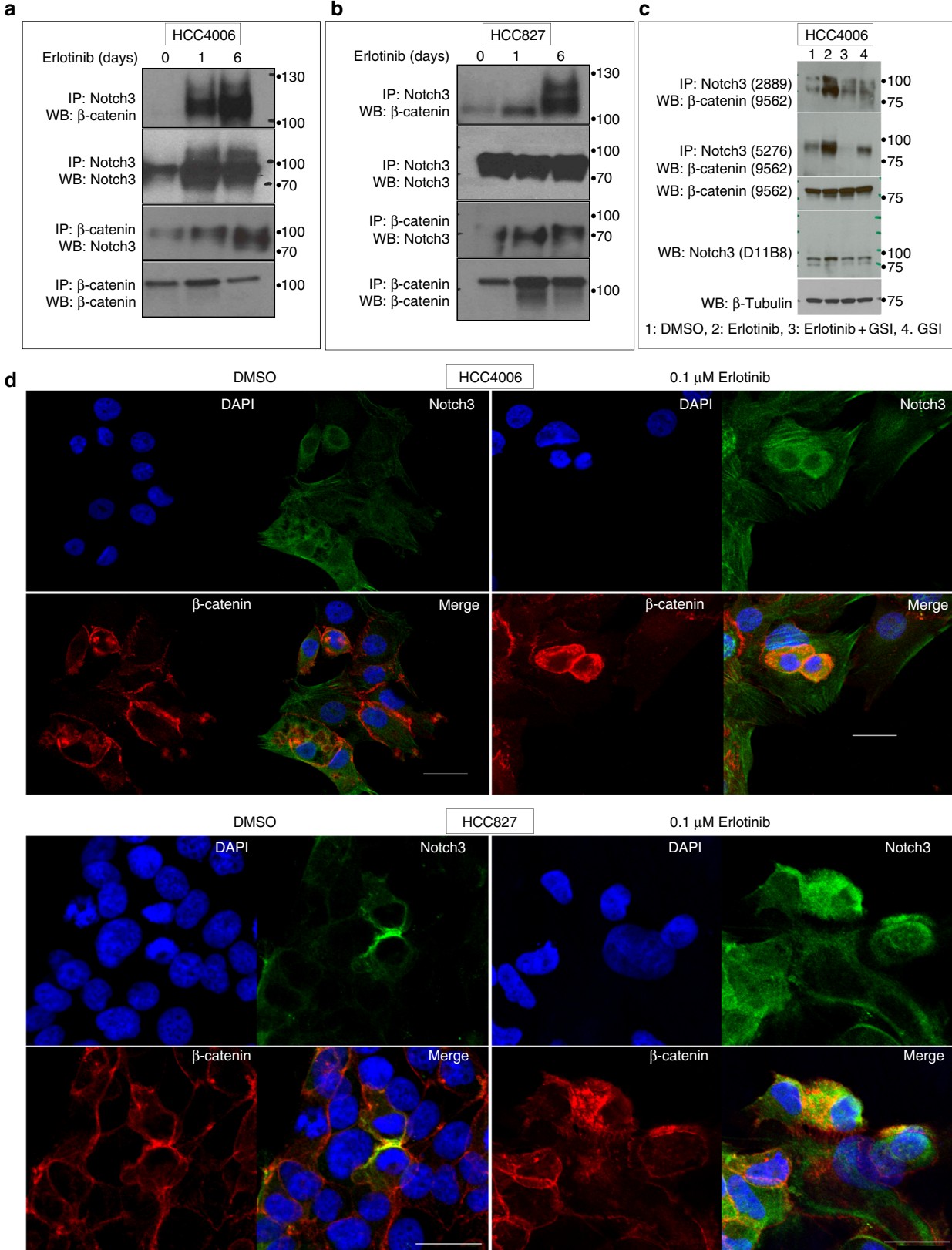

**DPCs with high β-catenin activity show stem-like phenotype.** To investigate how increased β-catenin activity might promote an EGFR-TKI persistent population, we assessed the protein levels of various stem cell markers in the population of treated cells with high β-catenin transcriptional activity. The EGFR mutant cell line HCC4006 was infected with a lentivirus expressing β-catenin reporter construct and treated with 0.1 μM erlotinib for 6 days, after which GFP$^{high}$ and GFP$^{low}$ cells were sorted by FACS. Lysates from GFP$^{high}$ and GFP$^{low}$ cells were analyzed by western blot using antibodies to total β-catenin and activated β-catenin.

**Fig. 3** Notch3 associates with β-catenin in an EGFR TKI-dependent manner. **a** HCC4006 and **b** HCC827 cells were treated with DMSO or 0.1 μM erlotinib for 1 or 6 days and cell lysates were immunoprecipitated (IP) with an antibody recognizing Notch3 and blotted (WB) with antibodies recognizing β-catenin or Notch3. A reciprocal IP with an anti-β-catenin antibody was performed and blotted with anti-Notch3 and anti-β-catenin antibodies. **c** HCC827 cells were treated with erlotinib or GSI alone or in combination and cell lysates were immunoprecipitated with Notch3 and blotted with β-catenin antibody. Cell lysates were also analyzed for Notch3, β-catenin, and β-tubulin. **d** HCC4006 and **e** HCC827 cells were treated for 6 days with vehicle (DMSO) or 0.1 μM erlotinib and stained with DAPI and probed with anti-Notch3 and anti-β-catenin antibodies. Images were overlayed to show co-localization. For (**d**) and (**e**) scale bar is 30 μm. Error bars represent SD from technical triplicates

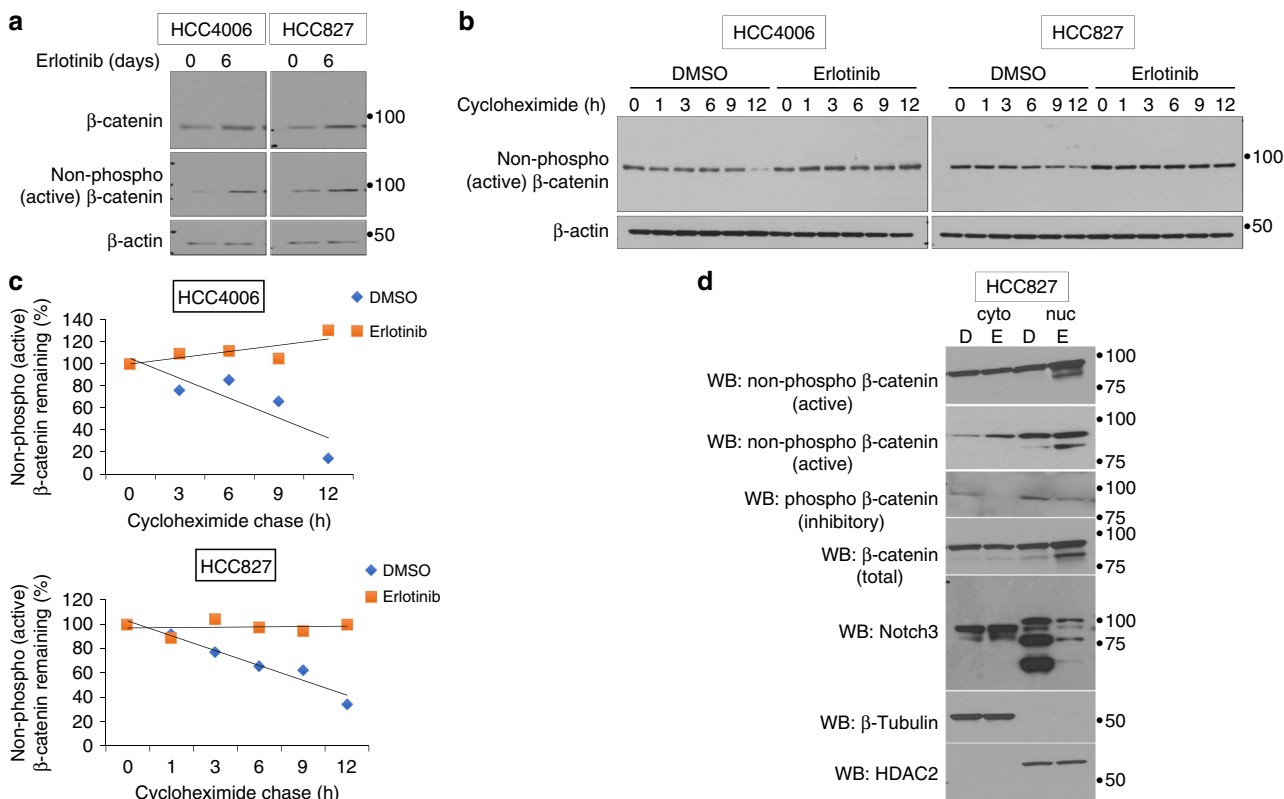

**Fig. 4** EGFR TKI treatment increase β-catenin protein stability. **a** Cells were treated as described in Fig. 3 and were analyzed for the expression of total and non-phospho (active) β-catenin by WB using an antibody that recognizes the active form of β-catenin. **b** Cells were treated as described in Fig. 4 and prior to harvesting, cells were further treated with cycloheximide for the indicated times. Cell lysates were analyzed for non-phospho (active) β-catenin by western blot. **c** Western blots were quantitated and data plotted (trend line) as percentage of non-phospho β-catenin remaining as a function of cycloheximide treatment. **d** HCC827 cells were treated with erlotinib or DMSO and subjected to cytoplasmic and nuclear fractionation. Equal amounts of total protein from each fraction was analyzed for β-catenin (active, inactive, and total), Notch3, β-tubulin, and HDAC2. Error bars represent SD from biological triplicates

As expected, GFP<sup>high</sup> cells showed an increase in total and active β-catenin (Fig. 5a). Cell lysates from GFP<sup>high</sup> and GFP<sup>low</sup> populations were analyzed for Notch3, as well as transcriptional targets of Notch and β-catenin, and for the expression of stem cell markers. In agreement with the immune fluorescence (IF) data, the western blots show increased levels of Notch3 protein in the fraction of cells enriched for β-catenin transcriptional activity. In addition, we see higher protein levels c-Myc and the stem cell markers Nanog and Oct4. The Notch transcriptional target Hey1 was not affected suggesting that transcriptional activation of canonical Notch3 targets is not necessary for DPC induction (Fig. 5a). These data do provide evidence that DPCs have stem-like properties due to activation of β-catenin and induction of β-catenin targets such as c-Myc and Nanog. To further demonstrate the functional role for EGFR-TKI-mediated β-catenin transcriptional activity, GFP<sup>high</sup> and GFP<sup>low</sup> fractions were subjected to

pulmosphere formation assays. Sphere formation of the GFP<sup>high</sup> cells was significantly greater than GFP<sup>low</sup> cells (Fig. 5b).

β-catenin pathway inhibitors, XAV939, a selective tankyrase-specific inhibitor of β-catenin-mediated transcription, and ICG-001, which inhibits interaction between β-catenin and its transcriptional coactivator CREB-binding protein (CBP), efficiently inhibited sphere formation in liquid culture (Fig. 5c). We used a limiting dilution assay (LDA) in immune compromised mice to assess in vivo clonogenicity. Cells were sorted based on GFP activity and injected into the flanks of mice at various dilutions. The GFP<sup>high</sup> fraction was much more potent in inducing tumors in mice than the GFP<sup>low</sup> cells suggesting that DPCs have enhanced clonogenicity (Fig. 5d). This suggests that β-catenin inhibitors, and specifically XAV939 and ICG-001 are compounds of potential utility for intervention studies to inhibit β-catenin activity and reduce EGFR TKI induction of DPCs[32–34].

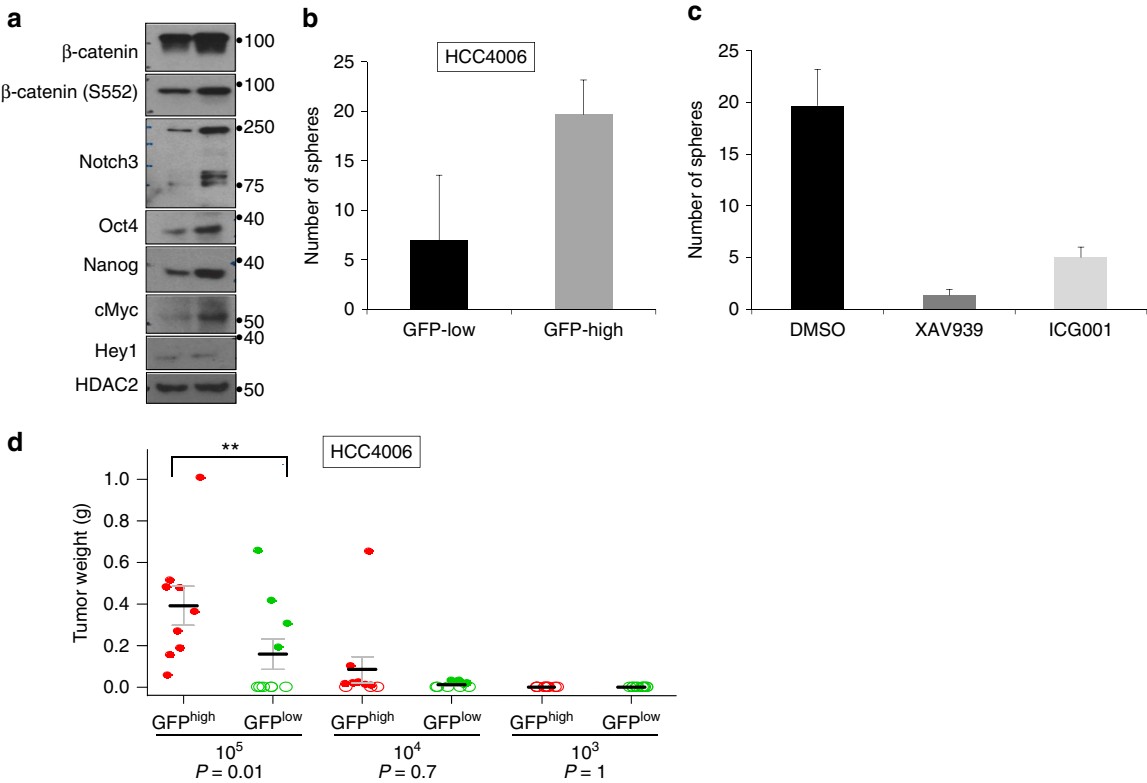

**Fig. 5** EGFR TKI treatment activates β-catenin signaling, which is responsible for the maintenance of DPCs. **a** HCC4006-Tcf/Lef-GFP reporter cells were treated with 0.1 μM erlotinib for 6 days and sorted for GFP expression by FACS into GFP$^{high}$ (high β-catenin activity) and GFP$^{low}$ (low β-catenin activity) cell populations, which were subsequently analyzed for Notch3, β-catenin, and stem cell markers by WB. **b** GFP$^{high}$ and GFP$^{low}$ cell populations were subjected to sphere formation assay. Quantification of total number of pulmospheres from both groups is shown. **c** EGFR TKI persistent pulmospheres are sensitive to β-catenin pathway inhibitors. HCC4006 cells were treated with erlotinib and TCF-GFP reporter positive cells were subjected to pulmosphere assay in the presence of DMSO or β-catenin inhibitors, XAV939, ICG-001. Quantification of total number of pulmospheres from all experimental conditions. **d** HCC4006-Tcf/Lef-GFP reporter cells were treated with 0.1 μM erlotinib for 6 days and cells with high and low GFP reporter activities were isolated by flow sorting. A limiting dilution assay was performed by injecting NSG mice with the number of GFP$^{high}$ and GFP$^{low}$ cells indicated. After 8 weeks the tumors were excised and tumor volumes and weights were measured. Solid-filled circles indicate individual tumor weights and open circles indicate no tumor grew at the site of injection. Red circles represent the group of tumors that were obtained by injection of GFP$^{high}$ HCC4006 cells. Green circles represent group of tumors that were obtained by injection of GFP$^{low}$ HCC4006 cells. Linear model was used to compare GFP$^{high}$ and GFP$^{low}$ groups within each group of cell dilution level. *P* values were adjusted for multiple comparisons by Holm's procedure. For (**b**) and (**c**) error bars represent SD from biological triplicates

**Induction of Notch3–β-catenin signaling in vivo**. To demonstrate the existence of EGFR TKI-mediated DPCs in vivo, HCC4006 and HCC827 xenografts tumors were treated with either vehicle control or erlotinib for 18 days. As expected, EGFR TKI treatment dramatically reduced the tumor size when compared to control mice. After 18 days of drug treatment, residual tumors from both HCC4006 and HCC827 xenografts were subjected to immunohistochemistry (IHC) staining to detect ALDH1A (Fig. 6a). Similar to what was seen in vitro, we observed a strong increase in ALDH1A staining after erlotinib treatment in both HCC4006 and HCC827 xenograft tumors. Furthermore, isolated tumor cells from HCC827 xenografts clearly showed an increase of ALDEFLUOR-positive cells upon erlotinib treatment (Fig. 6b). It is believed that the therapeutic resistance-mediated stem cell like phenotype is associated with the epithelial-to-mesenchymal transition (EMT) phenotype. To understand the link between drug persisters and EMT, we have performed IHC analysis of EMT markers in vivo using xenograft tumor samples that were treated with control or EGFR TKI and demonstrated that EGFR TKI leads to upregulation of the EMT phenotype (Fig. 6c–e). We have previously shown that Notch3 is involved in maintenance of the drug persistent population of cells and our in vitro data show that the protein levels of Notch3 and β-catenin

increase[9]. Therefore, we sought to determine the in vivo association between EGFR-TKI treatment and expression of Notch3 and β-catenin. Compared to control, EGFR TKI treatment significantly induced Notch3 protein expression, as well as total and activated β-catenin protein levels in xenograft tumors after treatment in vivo (Fig. 6f, g, h). Overall, the in vivo data supports our in vitro findings and suggests that tumor persistence after short EGFR TKI exposure is associated with induction of Notch3 and β-catenin. Co-localization studies supported our in vitro studies, showing co-localization of Notch3 and β-catenin and increased levels of β-catenin (Fig. 7a, b). In conclusion, our results demonstrate that EGFR TKI therapy leads to β-catenin and Notch3 induction in vivo as observed in vitro, associated with an increased clonogenicity.

**Notch3–β-catenin is increased in human tumor samples at the time of clinical resistance**. To determine whether the induction of Notch3 and β-catenin signaling is observed in human EGFR mutant NSCLC, we evaluated Notch3 and β-catenin levels in human lung cancer biopsies before and after erlotinib therapy. We performed IHC analysis of Notch3 and β-catenin on a total of three paired tumor samples obtained prior to the start of erlotinib

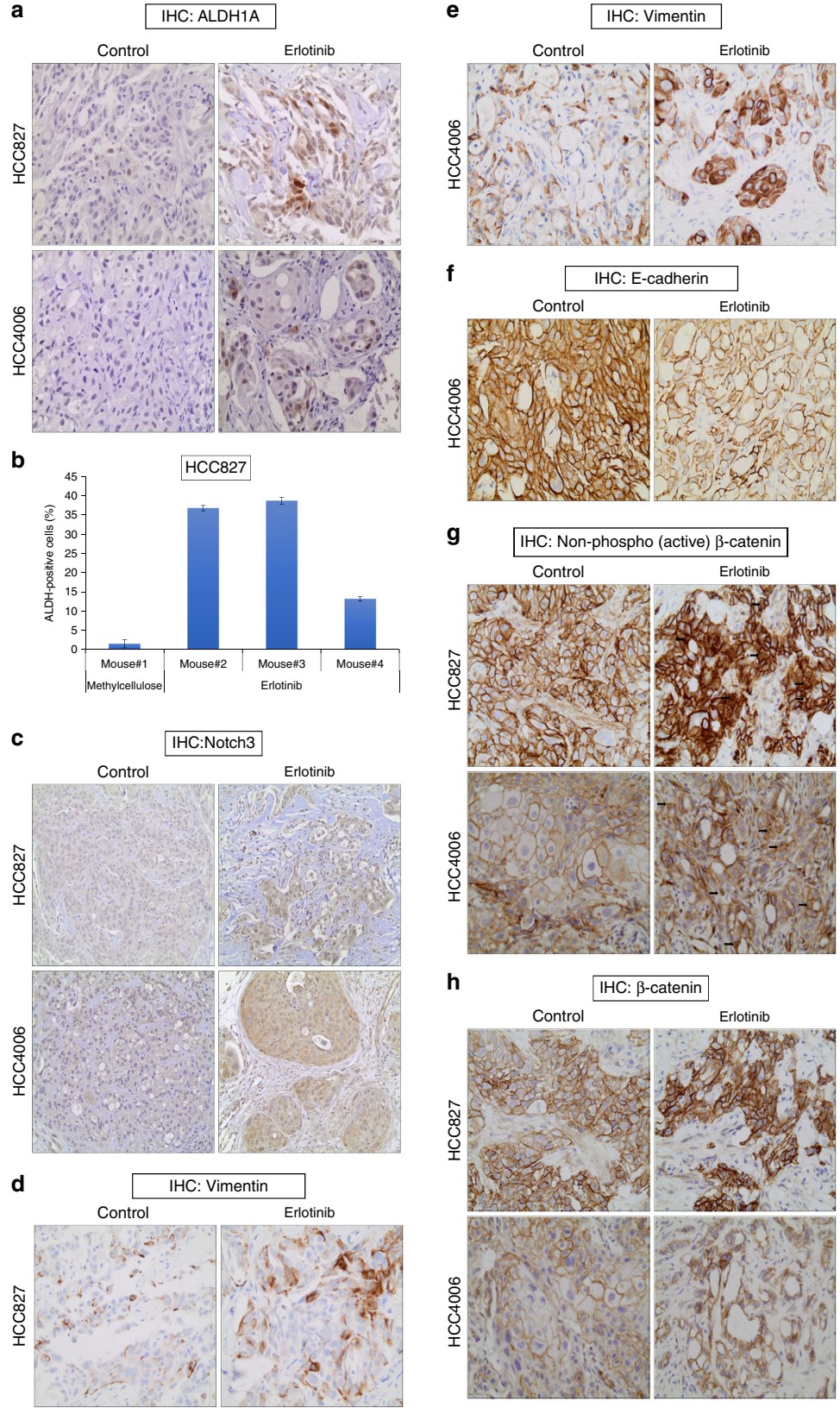

treatment and at the time of progression. A pathologist blindly evaluated these samples. Tumor biopsies taken at the time of progression showed increased levels of β-catenin expression (Fig. 8a). We also found increased Notch3 protein levels in these tumors (Fig. 8b). Taken together, these data suggest that our murine xenograft observations of induction of Notch3–β-catenin

expression in EGFR mutant NSCLC is also found in human tumors after EGFR TKI therapy.

**In vivo relevance of β-catenin inhibitors to EGFR mutant NSCLC.** After showing that EGFR TKI treatment increases β-catenin activity in in vitro and β-catenin protein levels in in vivo,

**Fig. 6** In vivo demonstration of EGFR TKI-induced drug persistent cells with EMT phenotype. **a** Mice with HCC827 (top) and HCC4006 (bottom) tumor xenografts were treated with methylcellulose (control) or erlotinib for 21 days. After the drug treatments tumors were harvested and subjected to IHC analysis for the putative stem cell marker, ALDH1A. **b** Mice with HCC827 tumor xenografts were treated with methylcellulose (control) or erlotinib for 21 days. Tumors were harvested and disrupted into single cell preparations, which were then subjected to ALDH assay. **c–e** EGFR TKI treatment increases Vimentin expression and decreases E-cadherin expression in vivo. **f** EGFR TKI treatment increases Notch3 expression in vivo. **g** EGFR TKI treatment increases the non-phospho (activated) β-catenin levels in vivo. **h** EGFR TKI treatment increases β-catenin levels in vivo. Error bars represent SD from techincal triplicates

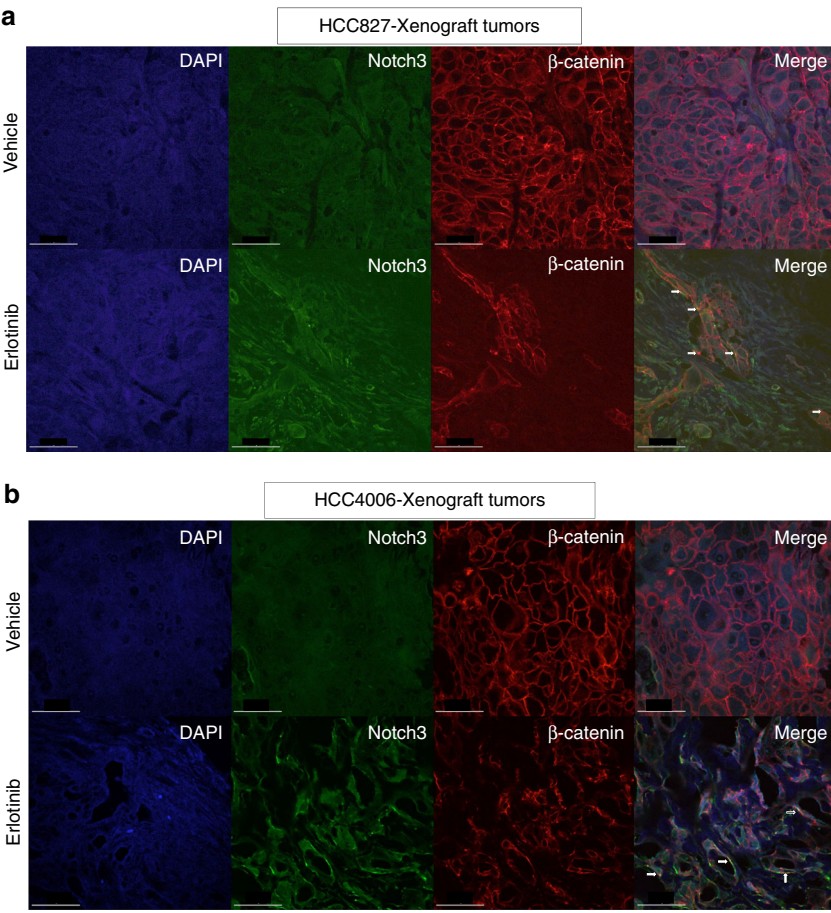

**Fig. 7** In vivo demonstration of EGFR TKI therapy-induced co-localization of Notch3 and β-catenin. **a, b** EGFR TKI treatment increases co-localization of Notch3 and β-catenin in vivo. Co-localization experiment was performed as in Fig. 3c and d. For (**a**) and (**b**) scale bar is 50 µm

we tested the effects of pharmacological inhibition of β-catenin using the clinical β-catenin pathway inhibitor ICG-001 in a xenograft model. ICG-001 is a small molecule inhibitor specifically blocks the interaction of β-catenin with the transcriptional coactivator CBP and inhibits β-catenin-mediated transcriptional activity[35–38]. To mimic the clinical management of EGFR TKI treatment, we treated mice with xenografts derived from the human EGFR mutant cell line HCC4006 with either erlotinib alone or in combination with ICG-001, and when tumors in the erlotinib group began to progress, treatments in both groups were stopped. Combined treatment with ICG-001 showed almost complete suppression of tumor growth (Fig. 9a). Survival analysis showed improved OS with combination treatment compared to erlotinib alone (Fig. 9b). Importantly, after stopping treatment we observed a significant delay in tumor recurrence and longer PFS in mice treated with the combination of erlotinib and ICG-001 compared to those treated with erlotinib alone (Fig. 9b). Treatment with ICG-001 alone did not show a significant anti-tumor

effect (Fig. 9a, b). We then treated HCC827 xenograft tumors with erlotinib alone or the combination of erlotinib and ICG-001 until the tumors were no longer apparent, then therapy was stopped and tumor recurrence was monitored. We observed a significant delay in tumor recurrence and longer PFS with mice treated with the combination of erlotinib and ICG-001 compared to those treated with erlotinib alone (Fig. 9c). Survival analysis showed a dramatic improvement of OS with the combination treatment (Fig. 9d). To demonstrate the specificity of the β-catenin inhibitor ICG-001, we have validated β-catenin transcriptional targets that were identified in this study (Supplementary Table 2), using HCC827 xenograft tumor samples that were treated with control, EGFR TKI alone or in combination with ICG-001. The gene expression data using qPCR analysis demonstrated that EGFR TKI treatment increases β-catenin transcriptional targets and further demonstrated that these β-catenin transcriptional targets were sensitive to the action of the ICG-001. This suggests that ICG-001 is able to inhibit the β-

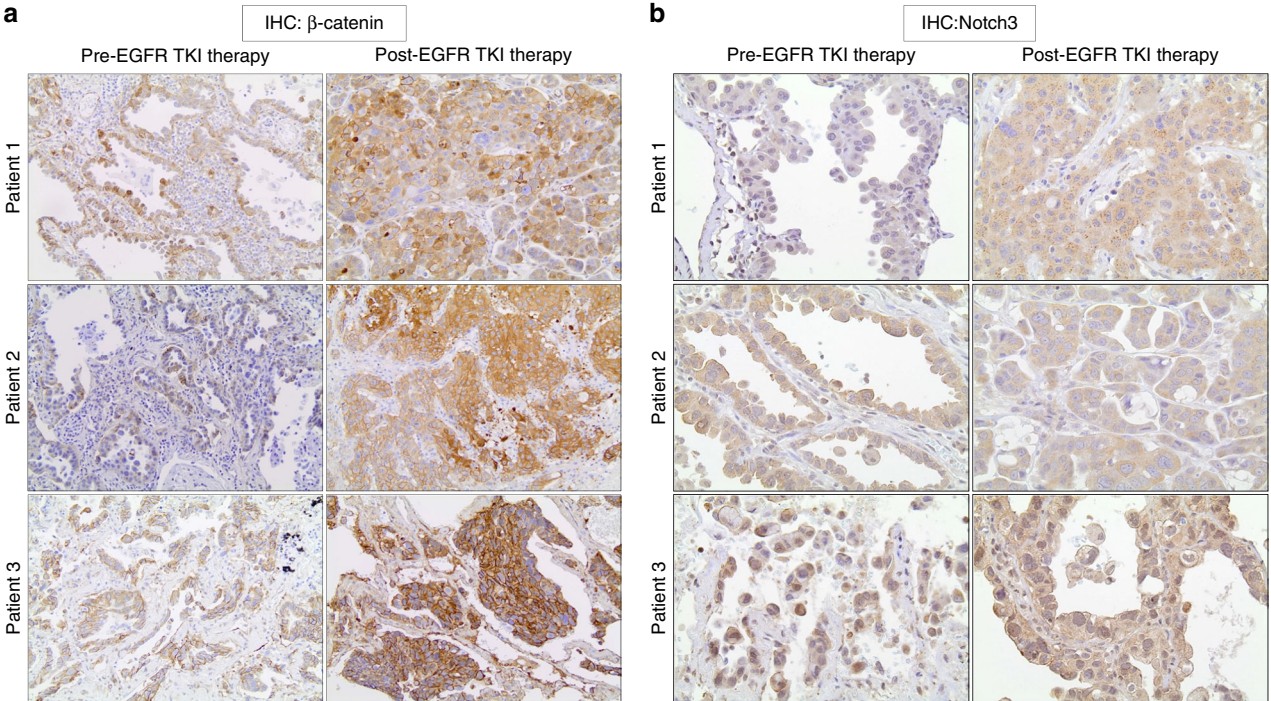

**Fig. 8** Increased Notch3 and β-catenin in patient biopsy samples at the time of acquired resistance to erlotinib. **a**, **b** Pre and post EGFR TKI therapy tumor biopsies were collected and IHC analysis of Notch3 (**a**) and β-catenin (**b**) was performed

catenin signaling in our pre-clinical studies (Supplementary Fig. 4). Taken together, these data indicate that targeting β-catenin signaling enhances EGFR-TKI efficacy leading to significantly longer PFS and OS in these EGFR mutant xenograft models.

**EGFR TKI-induced DPCs in the development of drug resistance.** To further understand the role of EGFR TKI-mediated β-catenin activation on persistence, we investigated the potential role of the transcriptional targets of β-catenin, which we initially identified in this study as biomarkers (Supplementary Table 1). Since PAI1 is a secreted protein, it may be useful as a non-invasive mechanism-related biomarker of baseline and induced β-catenin activation in lung tumors. To monitor the EGFR-TKI-dependent regulation of PAI1, we measured the secretion of this protein into the cell culture supernatant. Cells were treated with DMSO or erlotinib and the cell culture supernatant analyzed by a PAI1-specific ELISA at various time points. Analysis of secreted PAI1 levels in the medium suggest that erlotinib treatment rapidly increases PAI1 protein levels in the medium in a time-dependent manner achieving maximum secretion at day 7 of treatment (Supplementary Fig. 5). Similarly, EGFR TKI treatment enhanced soluble PAI1 levels in H358, H322, and H3255 cells (Supplementary Fig. 6a–c). We used multiple siRNAs targeting Notch3 and demonstrated that EGFR-TKI-mediated expression of PAI1 is eliminated by knockdown of Notch3 (Supplementary Fig. 7). To further demonstrate that EGFR TKI-mediated PAI1 expression is β-catenin dependent, HCC827 and HCC4006 cells were treated with EGFR TKI or ICG-001 alone or in combination, which showed that EGFR TKI-mediated PAI1 expression was sensitive to the ICG-001 (Fig. 10 a, b). These data suggested to us that PAI1 could be tested as a biomarker of β-catenin activation in vivo.

To assess whether we could observe induction of PAI1 in the serum of patients with EGFR mutant tumors undergoing EGFR TKI treatment, we performed a retrospective study in NSCLC

patients ($n = 42$) with EGFR-mutant tumors. Time to progression ranged from 21 days to 40.2 months. We used an ELISA to detect PAI1 protein in pre-treatment and post-treatment serum samples as a mechanism-based indicator for β-catenin activation. We observed a broad range in pre-treatment PAI1 levels, spanning from 0.8 to 134 ng/ml, suggesting a variable baseline β-catenin activation (Fig. 10c), at least some of which may be tumor-derived. We further wanted to determine whether EGFR TKI therapy could induce PAI1 levels and determine whether pre-treatment PAI1 levels could serve as the predictor for more rapid development of resistance. We found that 17 out of 42 patients (40%) that developed acquired resistance to EGFR-TKI had increased soluble PAI1 protein in their post-treatment serum compared to before therapy (Fig. 10c). Total PAI1 levels in the post-treatment group averaged 54 ng/ml compared to a pretreatment average of 35 ng/ml ($P = 0.004$). Interestingly, the patients with the lowest pre-treatment PAI1 levels showed the highest fold EGFR-TKI-mediated induction (Fig. 10d). When we dichotomized patients based on their baseline PAI1 levels into low and high groups based on a 15 ng/ml cutoff and performed a survival analysis using the Kaplan–Meier method, we found patients with low basal PAI1 levels had a significantly shorter PFS compared to patients with high basal levels with EGFR TKI treatment ($P = 0.02$ and HR = 0.48) (Fig. 10e). If we compared patients with a ≥2-fold induction to those with less than two-fold, there was a similar trend for worsened survival in those who showed significant induction of PAI1 (Supplementary Fig. 8). Overall, we conclude that PAI1 expression is a candidate marker for β-catenin activation and the development of drug persisters; thus, it is a potential surrogate marker for poor outcome in patients receiving EGFR TKI therapy and a potential biomarker for selecting patients for β-catenin-targeted therapy.

## Discussion

Tumor cells must persist following targeted therapy to enable the development of acquired resistance[39–47]. While NSCLC tumors

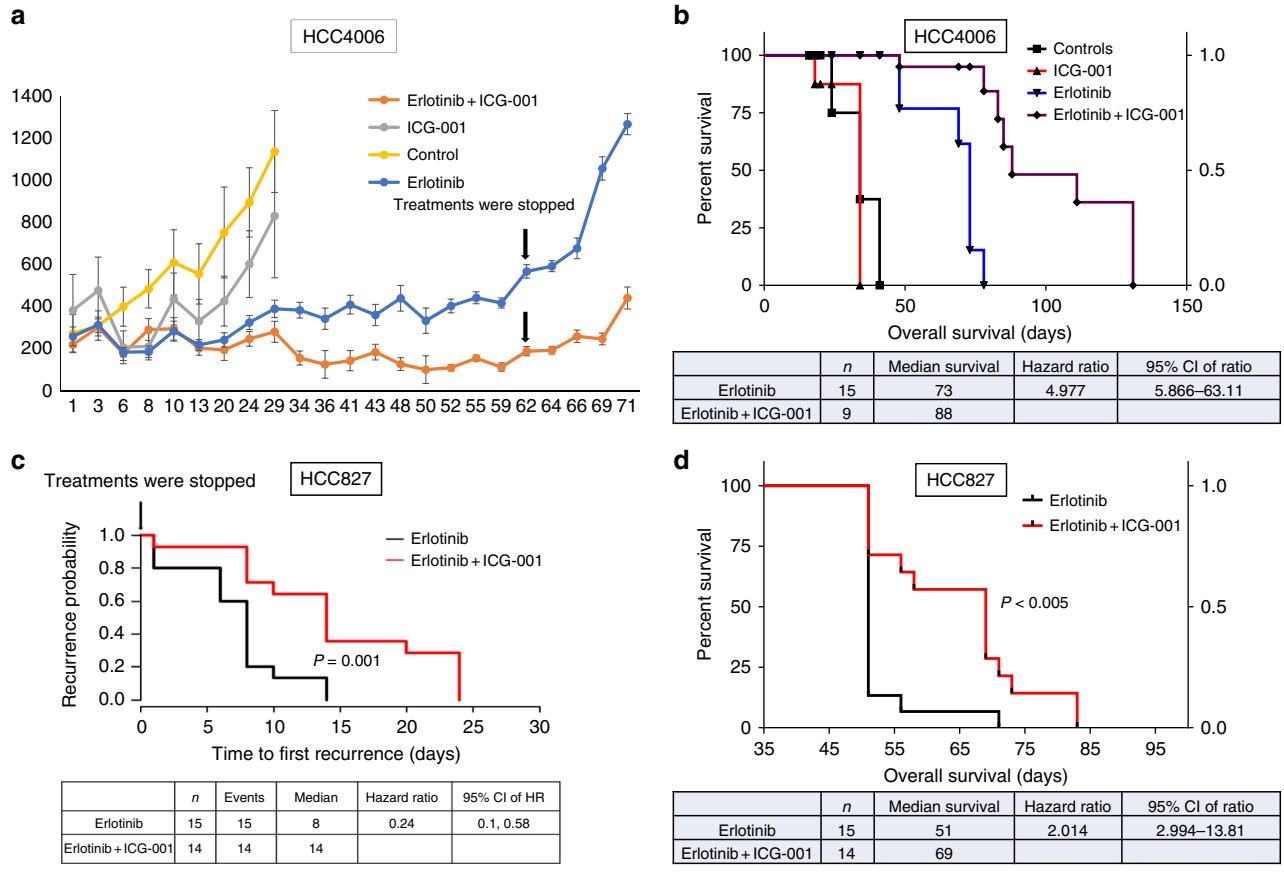

**Fig. 9** EGFR TKI treatment in combination with β-catenin inhibition strongly attenuates tumor onset in EGFR mutant NSCLC in vivo. **a** Human HCC4006 subcutaneous xenografts treated with erlotinib (50 mg/kg/day) or ICG-001 (150 mg/kg/day) alone or in combination for 5 days in a week for 9 weeks. **b** Overall survival analysis of HCC4006 xenograft mice that were treated with erlotinib or ICG-001 alone or in combination ($P = 0.0001$). **c** Human HCC827 subcutaneous xenografts were grown in mice that were treated with erlotinib (50 mg/kg/day) alone or in combination with ICG-001 (150 mg/kg/day) for 5 days a week for 3 weeks. After treatments were stopped, tumor recurrence was compared between the two groups. **d** Overall survival analysis of HCC827 xenograft mice that were treated with erlotinib alone or in combination with ICG-001 ($P = 0.0001$) for 21 days. Overall survival was determined after treatments were stopped. For (**a**) error bars represent SEM

with EGFR mutations usually show striking responses to EGFR TKIs, there is a wide range of response depth and duration between tumors with identical EGFR mutations[48]. Some tumors clearly have pre-existing resistant clones expressing T790M, but most have no such detectable clones and likely acquire resistance through acquired mutations. Until now, the signaling pathways responsible for rapid adaptive persistence in spite of active targeted therapy were not clear. Our data suggest that EGFR TKI therapy induces non-canonical Notch3-dependent activation of β-catenin that in turn leads to the survival of drug persistent clonogenic cells. Although cross-talk between Notch1 and Wnt/β-catenin signaling has previously been implicated in various cancers, the specific involvement of Notch3 in regulating β-catenin activity in EGFR-mutant NSCLC has not been reported, and the therapeutic benefit of direct β-catenin targeting in the context of EGFR TKIs have not been previously shown[49,50].

The opposing direction of effects of Notch1 and Notch3 in this effect is also interesting. In mouse embryos, for example, Notch1 ablation in Islet1 positive cardiac progenitor cells (CPCs) leads to proliferation of CPCs with increased levels of active β-catenin[14]. We also demonstrate that Notch1 depletion increases EGFR TKI-mediated β-catenin activation and DPCs. In contrast, knocking down Notch3 decreases β-catenin activation and depletes the number of drug persisters, suggesting that Notch1 and Notch3 act in an antagonistic manner. Lung cancers express variable levels of Notch family members and ligands and the complicated interplay

between the different Notch receptor and ligand family members in different tumors may underlie the observed variable in vitro efficacy of pan-Notch inhibitors such as GSIs, and the fact that the clinical trials using pan-Notch inhibitors with EGFR TKIs in lung cancer were not successful.

The mechanisms of β-catenin activation, and specifically that for Notch3 are poorly understood[51]. EGFR is known to regulate β-catenin signaling during development and in various cancers, but the effects of EGFR activity on β-catenin stability and activation have not been reported[51–53]. In this study, we have discovered a novel, potentially clinically significant, EGFR-TKI-induced association and co-localization of Notch3 and β-catenin associated with the stability of total and activated forms of β-catenin. We demonstrate that increased β-catenin activity leads to increased tumor take in xenograft models, suggesting that clonogenic DPCs survive through mechanisms that have CSC features. Collectively, these data establish a new understanding of the functional interaction between EGFR, Notch3, and β-catenin that is potentially particularly clinically important and feasibly targetable in human NSCLC tumors driven by constitutively activated EGFR. In these tumors, the loss of this potent EGFR signal induces Notch3 and β-catenin activation that in turn promotes the early persistence of DPCs in the presence of EGFR TKI.

Unlike Notch1, Notch3 is a weak transcriptional activator that lacks a strong transcriptional activation domain[54,55]. This indicates a potentially transcription-independent (non-canonical)

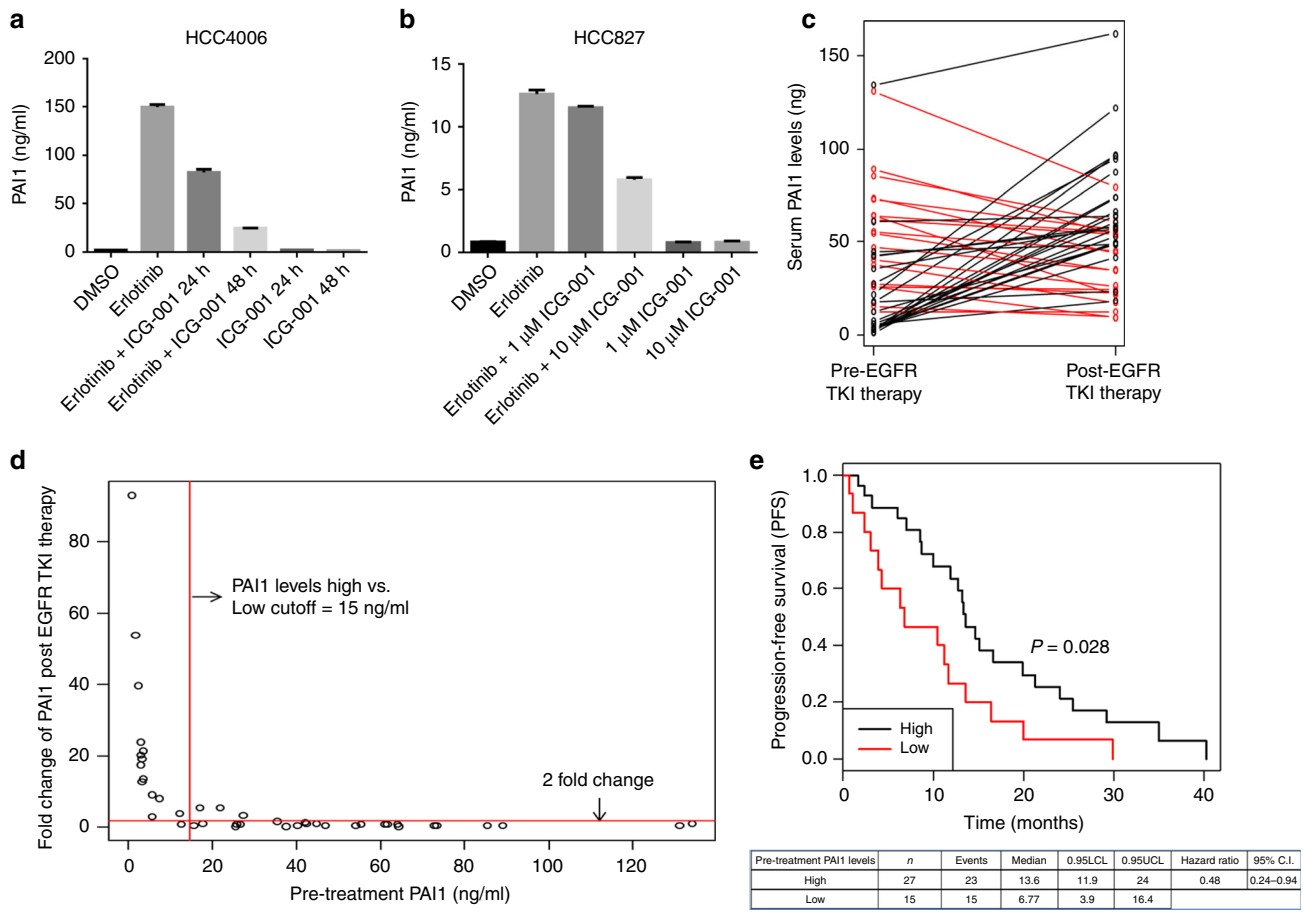

**Fig. 10** EGFR TKI therapy induces secretion of PAI1 in patient serum and predicts EGFR TKI resistance in EGFR mutant lung cancer. **a**, **b** HCC4006 and HCC827 cells were treated with erlotinib or ICG-001 alone or in combination for indicated time points with indicated doses and soluble PAI1 was measured from cell culture supernatant using PAI1 ELISA. **c** The serum PAI1 expression before and after EGFR TKI therapy in EGFR mutant NSCLC patients was compared by paired $t$ test. Line graph shows PAI-1 expression was significantly increased after treatment (54.36 ± 33.89 vs. 35.29 ± 30.81, $P = 0.004$). Each line represents one patient, connecting the serum PAI1 levels before and after EGFR TKI therapy. Black lines represent the patients who showed increased PAI1 levels after therapy while red lines represent those who do not show change or decreased PAI1 levels after therapy. **d**, **e** Patients expressing low levels of pre-treatment PAI1 show an increase in PAI1 levels after EGFR TKI therapy and have a decreased progression free-survival (PFS). **d** EGFR TKI-induced PAI1 increase in fold changes was plotted against basal PAI1 (ng/ml) levels. Red line parallel to $x$-axis indicates a two-fold change cutoff in the EGFR TKI therapy-induced PAI1 levels. Red line parallel to $y$-axis indicates PAI1 levels cutoff (15 ng/ml). **e** Kaplan–Meier method was used to estimate the progression-free survival (PFS) functions and log-rank test was used to compare PFS of the high and low pre-treatment PAI1 levels. Pre-treatment PAI1 level of 15 ng/ml was used as a cutoff. For (**a**) and (**b**) error bars are SEM

role for Notch3[15,16], which has been elusive or ignored in many previous studies. Using DN-MAML1 to block canonical activity, we show that the observed EGFR TKI induction of DPCs is independent of this canonical transcription pathway and the result of non-canonical specific Notch3 activation of β-catenin. Furthermore, our identification of β-catenin transcriptional targets induced by this Notch3 activity, and not by Notch1, reveals a novel aspect of Notch3 biology and points to the uniqueness of the Notch3 receptor and its selective importance in mediating drug-resistant persisters. Both the physical interactions between Notch3 and β-catenin revealed by co-immunoprecipitation and co-localization in vitro and the upregulation of Notch3 and β-catenin levels in patient tumor samples after acquiring resistance to EGFR TKI suggests that it may be direct association between Notch3 and β-catenin proteins that is responsible for the activation of β-catenin. Much needs to be done to better define the exact mechanism of these interactions.

Clinical resistance to EGFR TKIs is most frequently associated with secondary mutations in the EGFR gene (T790M, C797S) that directly affect drug binding and activity reactivating EGFR signaling, and activation of bypass pathways (AXL, HGF, Hh, IGFR, Met)[9,21,56–58]. It is clear that tumor heterogeneity and clonal selection of pre-existing resistant cells with EGFR T790M mutations is one mechanism for the development of resistance[56]. However, in other tumors, resistance to EGFR TKI appears not to be due to selection of pre-existing clones but the persistence of drug tolerant cells that survive long enough to acquire resistance de novo[10,59], including T790M. With the advent of third generation TKIs, bypass mechanisms and acquired C797S mutations will become more prominent. It should be emphasized that the findings reported here are also seen with these newer inhibitors, as it is a physiologic response inhibition of an activated EGFR driver.

We also show that induction of serum PAI1 in human patients with EGFR mutant tumors treated with TKIs is associated with shorter progression-free survival. Identification of a biomarker that is mechanistically associated with both innate and induced resistance to EGFR TKI therapy could be important for the

stratification of patients for studies, aid in early determination of patients who would develop resistance, and potentially define patients most likely to benefit from specific β-catenin-targeted therapeutic approaches to modulate this effect. Future studies will be required to determine if PAI1 expression directly affects tumor behavior in these patients[60–62].

Wnt/β-catenin pathway inhibitors have been demonstrated to show activity against stem-like cells and progenitors in various types of human cancers[36,37,63]. Biologic agents and small molecules targeting Wnt/β-catenin activity are active and well tolerated in clinical trials in other indications[36,49,64,65]. In this study, we demonstrated that selective β-catenin inhibitors, ICG-001 and XAV939, blocked the EGFR TKI-induced drug persisters in pulmosphere formation assays and improved outcomes in xenograft models with EGFR mutant tumors. This suggests potential clinical utility of these inhibitors targeting β-catenin as essential to successfully eliminating EGFR TKI-induced DPCs with the potential to extend the survival of patients with EGFR mutant NSCLCs. Our studies have demonstrated that the combination of ICG-001 and erlotinib resulted in: (1) a higher rate of complete tumor response, (2) a significantly prolonged time until tumor recurrence, and (3) an improved overall survival rate. The anti-stem cell and anti-tumor activity of ICG-001 in combination with erlotinib in these EGFR mutant NSCLC models is well tolerated and a very effective EGFR TKI combination in our studies. PRI-724, a more potent second-generation derivative of ICG-001 is currently in phase II clinical trials in other indications and could be combined with erlotinib or later generation EGFR TKIs in clinical trials for lung cancer.

In summary, the ability to survive in the presence of EGFR TKIs is clearly essential for the acquisition of clinical resistance. While the mechanisms of acquired resistance have been exhaustively studied, the mechanisms of adaptive persistence and potentially synergistic therapeutic strategies targeting this persistence in humans have been less well studied. We find that inhibition of EGFR potently induces β-catenin via a noncanonical Notch3-dependent mechanism and that this induction promotes the development of drug persisters and the early development of clinical resistance in animal models. Dual targeting of EGFR and β-catenin may be an effective clinical strategy, and serum PAI1 levels may be a useful mechanism-related marker of β-catenin activity in these patients.

## Methods

**Cell lines and transfections**. HCC4006, HCC827 and H358 cells were maintained in RPMI with 10% fetal bovine serum and were obtained from ATCC. siRNA transfections were performed with Lipofectamine RNAiMax (Invitrogen) reagent according to the manufacturer's instructions.

**Reagents**. PAI1 and MMP7 ELISA kits were purchased from R & D Systems. EGFR TKI, erlotinib was purchased from Chemietek and β-catenin pathway inhibitors, XAV-939, ICG001 were purchased from Selleck Chemicals.

**Western blot analysis and antibodies**. Western blot analysis was performed following standard procedures. Equal amount of protein was separated on 7.5% SDS–PAGE and transferred onto PVDF membrane. Western signals were detected using SuperSignal chemiluminescent substrate (West-Pico or West Dura Kits, Thermo Scientific). All of the uncropped western blots with molecular weight indicated were presented in Supplementary Figs. 9–14. Following antibodies were used in this study: β-catenin (1:1000, 8480 and 9562) phospho-β-catenin (1:1000 dilution, S552), Non-phospho (active) β-catenin (1:1000, 8814 and 4270), Notch3 (1:1000, D11B8), Notch3 (1:1000, 8G5), β-actin total (1:1000), Oct4 (1:1000, 2750), Nanog (1:1000, 4903) were purchased from Cell Signaling Technologies. Hey1 was purchased from Themofisher Scientific (1:1000, PA5-23484). cMyc antibodies were obtained from Sigma (1:1000, M-4439). HDAC2 antibody was purchased from Santa Cruz Biotechnology (1:1000, SC-6296).

**Plasmid constructs**. TOP flash/FOP flash reporter system was utilized for the detection of β-catenin transcriptional activity in HCC4006 cells. The TOP-flash reporter construct contains seven copies of wild type TCF/LEF-binding sites upstream of a thymidine kinase minimal promoter, which induces the transcription of GFP reporter. FOP-flash contains seven copies of mutant TCF/LEF-binding sites, which is not activated by β-catenin thus used as a negative control. TOP/FOP reporters were cloned into lentiviral vectors.

**Aldefluor assay and flow cytometry**. The Aldefluor assay kit (Stem cell Technologies) was used to determine the ALDH+ cells. The assay was performed according to manufacturer's instructions with modifications. Cells were suspended in aldefluor assay buffer and divided into two groups. One group was pretreated for 10 min with ALDH-specific inhibitor diethylaminobenzaldehyde (DEAB) before incubation with ALDH enzyme substrate bodipy-aminoacetaldehyde (BAA) for 45 min at 37 °C. Cells were spun down and re-suspended in a fresh aldefluor assay buffer to remove the unutilized substrate. Cells were analyzed on a FACSCalibur (BD Biosciences) Flow Cytometer. For the analysis of ALDH+ cells, DEAB-treated sample was used as a negative control and ALDH activity in presence of DEAB was considered as a baseline.

**siRNA transfections**. Silencer select siRNAs targeting non-targeting control (NTC#2) and silencer select siRNAs against Notch1 and Notch3 genes were purchased from ThermoFisher Scientific. HCC4006 cells were reverse transfected using 5 nM of each siRNA in an 800 µl of OptiMEM and 20 µl of LipofectamineRNAiMax (Invitrogen). Non targeting control #2: Cat. No.: 4390846 Notch1 (s453559) GCCUGGACAAGAUCAAUGATT UCAUUGAUCUUGUCCAGGCAG and Notch 3: –GGUGUGAACUGCGAAGUGATT–UCACUUCGCAGUUC ACACCTG.

Total RNA was isolated using miRNeasy kit (Qiagen). Medium was removed and 700 µl of QIAzol lysis reagent was used to lyse the cells and total RNA was isolated according to manufactures instructions.

**Mice and drug treatments**. All animal experiments were approved by The Ohio State University, Institutional Animal Care and Use Committee (IACUC). NOD/SCID mice were subcutaneously injected with HCC827 or HCC4006 cells ($2 \times 10^6$) suspended in 100 µl PBS into the flanks. Once the tumors reached palpable size, they were treated with erlotinib (50 mg/kg) or ICG-001 (150 mg/kg) or in combination with various time periods. Erlotinib was administered through oral gavage at 50 mg/kg/day for 5 days in a week. ICG-001 was administered through intra peritoneal injection at 150 mg/kg/day for 5 days in a week. Tumor sizes were serially measured with calipers and mouse weights were monitored three times a week throughout the study. Tumor volume was calculated using the formula: tumor volume = length/2 × width) 2/2 reported as the mean ± SEM.

**Limiting dilution assay**. HCC4006 cells stably expressing Tcf/Lef-GFP reporter cells were treated with erlotinib for 6 days and GFP$^{high}$ and GFP$^{low}$ population of cells were FACS sorted. To demonstrate the role of β-catenin transcriptional activity in EGFR TKI-mediated cancer stem cell phenotype LDA was performed. Various number of GFP$^{high}$ and GFP$^{low}$ population of cells were subcutaneously injected into flanks of NSG mice. Tumor appearance was measured weekly for 8 weeks.

**Co-immunoprecipitation, immunoprecipitation, and western blotting**. Cells were washed twice in ice-cold phosphate buffered saline and for co-immunoprecipitation experiments harvested and lysed with NP40 lysis buffer (10 mM phosphate buffer, 120 mM NaCl, 2.7 mM KCl, 1% Nonidet P40, 10% glycerol) or for western analysis, cells were lysed with RIPA buffer (10 mM phosphate buffer, 120 mM NaCl, 2.7 mM KCl, 1% Nonidet P-40, 0.5% DOC, 0.1% SDS) supplemented with complete mini-EDTA-free protease inhibitor mixture (Roche) and phosphatase inhibitor mixture cocktails 2 and 3 (sigma), 2 mM NaF and pervanadate for immunoprecipitation for detection of phosphorylation. Co-immunoprecipitations were performed using 2 mg of protein lysate precipitated with Notch3 (D11B8) Rabbit mAb (Cell Signaling Technology) or β-catenin (D10A8) Rabbit mAb (Cell Signaling Technology). Antibody–Protein complexes were isolated using protein G-coupled magnetic beads and equal amounts of immune precipitate was separated on SDS-PAGE and western analysis was performed to detect Notch3 or β-catenin. For western analysis, protein lysates were quantified by using Bio-Rad protein assay kit and equal amount of proteins were separated on 7.5% SDS–PAGE and subjected to western blot analysis.

**Pulmosphere formation**. HCC4006 cells were treated with erlotinib for 6 days and M50$^{high}$ and M50$^{low}$ cells were isolated by FAC sorting. HCC4006 M50$^{high}$ and M50$^{low}$cells were seeded in a 24-well ultra-low attachment plates in a DMEM/F-12 knockout medium supplemented with 20% knockout serum replacement, L-glutamine, penn/strep, basic FGF (10 ng/ml) and EGF (29 ng/ml). The resultant cell spheres were dissociated with TrypLE (Life Technologies) and serially diluted for single spherical cell formation. The second generation of single spherical cells was expanded for all further experiments. The numbers of spherical colonies were counted. For the treatment studies, XAV939, ICG-001 were added to the pulmosphere culture.

**Immunofluorescence (IF) staining of cells**. HCC827 and HCC4006 cells were seeded in eight-well chamber slides and treated with vehicle, DMSO or erlotinib for 6 days. Cells were fixed with 4% paraformaldehyde and incubated with primary antibodies against Notch3 rabbit polyclonal antibody (Allele biotechnology, ABP-PAB-10683), β-catenin (L54E2) Mouse mAb-Alexa Fluor 555 conjugate (Cell Signaling Technology, 2677) alone or together overnight at 4 °C. After washing, cells were stained with Alexa Fluor- 488 or 594 conjugated secondary IgG antibodies. To detect the nuclei, the slides were mounted with Vectashield anti-fade Mounting medium with DAPI (Vector Laboratories, Inc., Burlingame, CA, USA). Cells were visualized using an Olympus FV1000 Filter confocal microscope fitted with a ×40 oil objective.

**IF staining of tumor xenografts**. Tumor samples were fixed with 4% paraformaldehyde, processed and embedded in paraffin. Standard immunohistochemical techniques were used according to the manufacturer's recommendations (Vector Laboratories) using antibodies against Notch3 (Allele biotechnology, ABP-PAB-10683), β-catenin (Cell Signaling Technology, 2677), avidin DH: biotinylated horseradish peroxidase H complex with 3,3′-diaminobenzidine (Polysciences), and Mayer's hematoxylin (Fisher Scientific). For IF staining, Notch3 primary antibodies and Alexa 488/568 secondary antibodies (Life Technologies, NY) were used for labeling. Xenograft tumor sections were de-waxed by heating on a slide warmer at 60 °C for 15 min.

**Immunohistochemistry**. IHC staining was performed in standard fashion on paraffin-embedded tissue. Embedded tissue was cut at 4-μm utilizing positively charged slides. Slides with sections were then placed in a 60 °C oven for 1 h, cooled, deparaffinized and rehydrated through xylenes and graded ethanol solutions to water. Slides were quenched for 5 min in a 3% hydrogen peroxide solution in water to block endogenous peroxidase. Slides then underwent heat-induced epitope retrieval employing Target Retrieval Solution (S1699, Dako, Carpinteria, CA) for 25 min at 96 °C in a vegetable steamer (Black & Decker), were cooled for 15 min, and then placed on a Dako Autostainer Immunostaining System. All incubations on the Autostainer were for 60 minutes at room temperature. Mach 3 Rabbit detection system (HRP Polymer, M3R531L, Biocare Medicals, Concord, CA) was applied for 20 min. Staining was visualized with the Liquid DAB + Chromogen (K346811, 5 min development, Dako, Carpinteria, CA). Slides were then counterstained in Richard Allen hematoxylin (Thermo Scientific, Middletown, VA), dehydrated through graded ethanol solutions, cleared in xylene and coverslipped. Notch3 antibody (Rabbit polyclonal, Allele Biotechnology, San Diego, CA) was used at 1:100 dilution. β-Catenin antibody (Mouse monoclonal, Clone 14, BD Biosciences, San Jose, CA) was used at 1:3500 dilution. Non-phospho (active) β-catenin antibody (Rabbit monoclonal, Cell Signaling Technology, Danvers, MA) was applied at 1:1000 dilution.

**Gene expression analysis**. Whole transcriptome analysis was done using Gene-Chip Human Transcriptome Array 2.0 from Affymetrix. For affymetrix gene expression data, signal intensities were quantified by affymetrix expression console. Raw probe intensities were processed with background correction and normalization, probeset-level summarization and noise-level gene filtering. Linear models were performed to detect differentially expressed genes employing a variance shrinking method. The mean number of false positives 5 out of 10,000 and fold change 2 were used together to identify the differentially expressed genes. TMeV 4.9 (Multi Experiment Viewer) was used to visualize top genes in the heatmap. Hierarchical clustering with average linkage method based on Euclidean distance metric was used for the clustering in the heatmap.

**Quantitative RT-PCR**. Total RNA was isolated using the miRNeasy Kit (Qiagen) according to the manufacturer's instructions and synthesis of cDNA from total RNA (100 ng) was performed using a commercially available kit (Applied Biosystems, Foster City, CA). Reverse transcription thermo cycling parameters were as follows: 25 °C for 10 min, 37 °C for 60 min, 37 °C for 60 min 85 °C for 5 min for mRNA; Reactions were performed on a MyCycler (Bio-Rad, CA, USA). Real-time PCR was performed using the ABI 7900HT system in 10 μl reaction volumes containing 5 μl of PCR master mix (Taqman 2x Universal PCR master mix, cat # 4324018) each primer probe 0.5 μl, 3.5 μl of nuclease-free water, and 1 μl of cDNA (total 10 μl) in optical 384-well plates (Applied Biosystems). Cycling conditions: 95 °C for 10 min, followed by 40 cycles of 95 °C for 15 s and 60 °C for 60 s. Triplicate qPCR reactions were performed for each cDNA sample for all experiments. The threshold fluorescence level was set manually for each plate using SDS software version 2.3 (Applied Biosystems). Following the export of Cycle threshold (Ct) data, further data analysis for both platforms was performed in Microsoft® Excel 2003. Comparison of slope and R2 values between pre-amplified and non-amplified cDNA, as a template on the BioMark arrays was performed using Student's paired t-test. Gene expression was verified using primer sets specific for PAI1 and MMP7.

**PAI1 and MMP7 ELISA assay**. Cells were treated with vehicle control (DMSO), erlotinib or GSI, PF for 6 days in growth medium. At the end of the treatment period, the cell culture medium was collected and centrifuged to remove cell debris. For each group cell number was determined. PAI1 and MMP7 ELISAs (R&D Systems) were performed according to manufacturer's instructions. The ELISA values were normalized to cell number.

**Patients**. All of the patients that were participated in this study were provided written informed consent at five participating institutes in Japan. This study was also approved by the Institutional Review Boards (IRB) of the each of the participating institutes. A total of 42 pre-EGFR-TKI and post-EGFR-TKI serum samples were obtained from 42 lung cancer patients with EGFR mutations. EGFR TKIs gefitinib and erlotinib were given to 29 and 13 patients, respectively. Exon 19 deletion of EGFR, and L858R point mutation in exon 21 of EGFR were detected in 16 and 24 of the tumors, respectively. Two patient tumors had neither exon 19 deletion nor L858R mutation; one patient's tumor had both G719A point mutation in exon 18 and L861Q point mutation in exon21 and the other patient's tumor had a L861Q point mutation in exon21. Pre-treatment serum samples were collected after the detection of EGFR mutation but before the administration of EGFR TKI. Post-treatment serum samples were collected within 7 days of determination of EGFR TKI-mediated acquired resistance.

**PAI1 detection in human serum samples**. Serum PAI1 levels were measured using PAI1 ELISA (Quantikine ELISA kits, R&D Systems) in accordance with the manufacturer's procedures. All samples were run as singlets. Color intensity was measured at 450 nm with a spectrophotometric plate reader. PAI1 concentrations were determined by comparison with standard curves. The detection limit for PAI-1 was 0.313 ng/ml. Serum PAI1 expression before, and after treatment was evaluated; changes in serum expression levels were used to separate patients into two groups and correlate clinical outcome. Patients with post-treatment serum PAI1 levels more than two-fold higher than pre-treatment were placed in the increased expression group; patients with post-treatment serum PAI1 levels less than two-fold higher than pre-treatment were placed in the no-change group.

**Notch stimulation**. HCC827 cells expressing GFP vector alone or DN-MAML1 were seeded at $1 \times 10^5$ cells/well in 12-well plate that were coated with PBS containing rhDLL1-Fc chimera (10 μg/ml) or PBS for overnight. Recombinant human DLL1-Fc chimera was purchased from R&D Systems (1818-DL). Cells were allowed to grow for 48 h and total RNA was isolated for Hes1 expression analysis.

**Fractionation of cells**. Cytoplasmic and nuclear fractionation of HCC827 cells was performed using NE-PER nuclear and cytoplasmic extraction reagents kit (Thermo scientific, 78833). Cells were treated with DMSO or erlotinib for 6 days prior to fractionation. At the time of the fractionation cell culture medium was removed and washed twice with ice cold-PBS. Cells were collected in CERI buffer and rest of the procedure was done according to manufacturer's instructions. Nuclear and cytoplasmic protein fractions were subjected to protein estimation and equal amount of protein was separated on 7.5% SDS-PAGE and subjected to western analysis to determine the active-β-catenin using two different antibodies (1:1000), phospho β-catenin (1:1000), total β-catenin (1:1000), Notch3 (1:1000), β-tubulin (1:1000), and HDAC2 (1:1000) levels.

**Statistical analysis for PFS of patients with EGFR NSCLC**. Progression-free survival was measured using the Kaplan–Meier method; survival outcomes were compared using the Wilcoxon test. This test was two-sided, and $p < 0.05$ was taken to indicate statistical significance. All statistical analyses were performed using GraphPad Prism Version 6.07 (GraphPad Software, Inc., San Diego, CA).

**Data availability**. The data that support the findings of this study are available from the authors upon reasonable request.

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

## Acknowledgements

We are grateful to Michael Koenig for critical reading of this manuscript. This work was supported by The DallaPezze Family Foundation and NIH/NCI RO1CA175370.

## Author contributions

R.R.A. conceived and designed the study. R.R.A. and D.P.C. oversaw the study. R.R.A., K.S., T.Y., J.Z., R.G., W.W., T.T., S.M., W.D., P.F., M.A.R., P.N.S., J.E., J.A., E.E.T. and M.M.D. performed experiments and analyzed data. T.Y., S.Y., S.T., K.F., N.K., K.T., F.O.and Y. N. provided clinical samples. R.R.A., D.P.C. and J.A. wrote the manuscript.

## Additional information

**Competing interests:** A patent application to the US patent office is pending. D.P.C. has done consulting for Abbvie, Adaptimmune, Agenus, Amgen, Ariad, AstraZeneca, Biocept, Boehringer Ingelheim, Bristol Myers-Squibb (BMS), Celgene, Foundation Medicine, Genentech/Roche, Gritstone, Guardant Health, Inovio, Merck, MSD, Novartis, Palobiofarma, Pfizer, prIME Oncology, Stemcentrx, Takeda, and has unrelated grant support from BMS. The remaining authors declare no competing interests.

