## [Peer Review File · Nature Communications]

Reviewers' comments:

Reviewer #1 (Remarks to the Author):

This excellent manuscript provides convincing evidence for a novel mechanism for acquired TKI resistance in NSCLC, based on a non-canonical activity of Notch3, which binds and stabilizes beta-catenin. Stem-like, TKI-resistant cells developing under TKI treatment show beta-catenin dependence and are sensitive to investigational beta catenin inhibitors. A beta catenin target gene, PAI1, is induced under these conditions in preclinical models as well as in TKI-treated patients, and plasma levels of PAI1 appear to have prognostic value. Overall, the manuscript presents very solid and highly translationally relevant observations. This function of Notch3 had not been described before. Importantly, Notch1 seems to have opposite effects in this model, although Notch1 had been previously reported to interact with beta catenin. Hence, this effect seems to be Notch3-selective. This manuscript could be further strengthened by addressing some remaining technical and mechanistic issues. Specifically:

- Figure 2F: The effect of DN-MAML appears to be incomplete, and significant levels of Notch-reporter activity remain even in stable DN-MAML transfected cells. DN-MAML blocks all Notch transcriptional complexes. Is the residual reporter activity non-specific? This could be assessed by determining the levels of endogenous Notch transcripts, including Notch3-selective transcripts such as HES5. Stable DN-MAML cells ought to show nearly complete suppression of such transcripts if the DN-MAML protein is expressed at sufficient levels to block Notch-dependent transcription.

- Figure 3: based on the apparent molecular mass, the Western blots presented could show either the transmembrane (uncleaved) or the intracellular (cleaved) form of Notch3. Binding of beta catenin to the uncleaved form of Notch1 has been reported in other systems. Unfortunately, reliable commercial antibodies specific to the cleaved form of Notch3 do not exist yet. However, if beta catenin binds to uncleaved Notch3, the association shown in this figure would be unaffected by gamma secretase inhibition, or perhaps even enhanced. Conversely, if the Notch3 cleavage is necessary for this association, gamma secretase inhibition would prevent it.

- A related question is how beta catenin is released to activate transcription. For instance, Notch3 activation by a ligand or in a ligand-independent fashion could trigger release of beta catenin into the nucleus. Alternatively, active beta-catenin and cleaved Notch3 could migrate into the nucleus together and accumulate at beta-catenin-responsive promoters. The confocal images shown in Figures 3C, D and 7A,B seem to indicate that both cytoplasmic and nuclear Notch3 are increased by erlotinib, suggesting that at least a fraction of the Notch3 is activated. However, most of the beta catenin signal seems to remain in the cytoplasm. Is there an increase in nuclear beta catenin under erlotinib treatment, as transcript analysis suggests, and is this paralleled by an increase in nuclear Notch3? This is hard to quantify from the images presented, but it could be assessed by re-analyzing the images or, better, by directly measuring cytoplasmic and nuclear beta catenin and Notch3.

- Minor points:

- o Line 100: there are numerous, well characterized non-canonical functions of Notch receptors in addition to interaction with beta catenin

- o Line 234: Myc is also a Notch transcriptional target, not an exclusive beta catenin target.

Reviewer #2 (Remarks to the Author):

This study describes a non-canonical activation of b-catenin signaling through Notch3 as a mechanism of adaptation to and resistance to EGFR TKI treatment in NSCLC. The manuscript is well-written and the experiments are clear. However, there are several issues, described below, that limit the strength and novelty of the findings at this stage.

1- The authors main novel findings in NSCLC are: (a) the interaction between b-catenin and Notch-3, and (b) the correlation between serum PAI1 level and EGFR TKI resistance.

Regarding the former, literature data show an opposite correlation between b-catenin activity and Notch-3 interaction in other cancer types such that Notch-3 inhibits b-catenin function. As the authors stated, Notch-3 stabilizes beta-catenin in the cytoplasm and indeed co-IF stains show cytoplasmic beta-catenin and Notch-3 (Figs. 3 C-D and 7 A-B). How do the authors reconcile their data with the literature and how do they link this finding to b-catenin function since it is expected that b-catenin function (transcriptional activity) requires its nuclear instead of cytoplasmic localization.

Regarding the latter, high PAI1 expression was detected in one cell line (HCC4006) post TKI. Do other cell lines increase PAI1 post-TKI in a beta-catenin dependent fashion? In fact serum PAI1 level is already high in a significant patient cohort pre-TKI (Fig. 10). These data do not support a specific role for rapid, TKI-induced increases in PAI1 (via b-catenin activation) in patients. Additionally, what other factors beyond b-catenin regulate PAI1? How specific is PAI1 expression as a biomarker of b-catenin activation during EGFR TKI treatment?

2- The authors state that activating mutations in the Wnt/b-catenin pathway are rare in NSCLC. Recent published data indicate that this may not be the case, particularly in the setting of EGFR TKI disease persistence and resistance (PMIDs: 28854272, 29106415). In this light, the authors main finding, while interesting, is not especially novel.

3- A central conclusion of this study is that combined treatment of EGFR TKI + b-catenin inhibitor is more effective than TKI monotherapy. Other papers have demonstrated the efficacy of this approach, as the authors mentioned. The novelty here is limited. Additionally, the effects of the b-catenin inhibitor alone are not shown in vivo nor is the molecular validation in tumors of its on-target effects and whether the effects can be rescued through constitutive activation of b-catenin (validating the on-target activity of the drug in these systems).

4- In general, the in vivo efficacy data show modest effect sizes. The magnitude of benefit of the combination therapy appears unlikely to correlate with significant improvements in the control of NSCLC in the clinic. It is also unclear how the in vivo data specifically demonstrate that it is induction of b-catenin versus selection for pre-existing cells with high b-catenin signaling during EGFR TKI treatment that leads to tumor progression over time.

5- b-catenin and EGFR TKI resistance have been linked to EMT. Have the authors examined this connection in their systems? Do the patient specimens show evidence of an EMT, particularly in association with EGFR TKI resistance and b-catenin activation.

Response to Reviewers' comments:

We sincerely would like to thank the reviewers for their comments and constructive criticisms regarding our manuscript titled “**Notch3-dependent β -catenin signaling mediates EGFR TKI drug persistence in EGFR mutant NSCLC**”. The revised manuscript has additional data requested by the reviewers and we have addressed the concerns raised by the reviewers in a detailed point-by-point response below. We feel that the new data significantly strengthens the manuscript and hope that you now find it satisfactory for publication in Nature Communications.

Response to reviewer 1

- Figure 2F: The effect of DN-MAML appears to be incomplete, and significant levels of Notch-reporter activity remain even in stable DN-MAML transfected cells. DN-MAML blocks all Notch transcriptional complexes. Is the residual reporter activity non-specific? This could be assessed by determining the levels of endogenous Notch transcripts, including Notch3-selective transcripts such as HES5. Stable DN-MAML cells ought to show nearly complete suppression of such transcripts if the DN-MAML protein is expressed at sufficient levels to block Notch-dependent transcription.

We apologize to the reviewer because a labeling mistake on our original figure had the panel labeled as “2F” when in actuality it was 2B. That has been corrected. While there is a significant drop in reporter activity from 3.9 to 1.7-fold in Figure 2B, there is residual reporter activity that could be non-specific. To strengthen our case the DN-MAML was working as intended, we performed additional analysis of HES1 expression in cells stimulated with Notch ligand, DLL4, which clearly showed that basal HES1 transcript levels in unstimulated cells are slightly lower with DN-MAML1. On the other hand, DN-MAML1 was almost completely abolished following ligand activated HES1 expression suggesting that DN-MAML1 is active in inhibiting canonical Notch signaling (**Fig. 2c**). As per the reviewer’s suggestion we also performed HES5 expression analysis. However, we did not detect HES5 expression in these cells.

- Figure 3: based on the apparent molecular mass, the Western blots presented could show either the transmembrane (uncleaved) or the intracellular (cleaved) form of Notch3. Binding of beta catenin to the uncleaved form of Notch1 has been reported in other systems. Unfortunately, reliable commercial antibodies specific to the cleaved form of Notch3 do not exist yet. However, if beta catenin binds to uncleaved Notch3, the association shown in this figure would be unaffected by gamma secretase inhibition, or perhaps even enhanced. Conversely, if the Notch3 cleavage is necessary for this association, gamma secretase inhibition would prevent it.

The co-immunoprecipitation experiment was done in cells treated with EGFR TKI, Gamma Secretase Inhibitor (GSI) alone or in combination and found that EGFR TKI mediated Notch3 association with β -catenin is sensitive to GSI suggesting that cleaved Notch3 (intracellular) could be associating with β -catenin (**Fig. 3c**).

- A related question is how beta catenin is released to activate transcription. For instance, Notch3 activation by a ligand or in a ligand-independent fashion could trigger release of beta catenin into the nucleus. Alternatively, active beta-catenin and cleaved Notch3 could migrate into the nucleus together and accumulate at beta-catenin-responsive promoters. The confocal images shown in Figures 3C, D and 7A,B seem to indicate that both cytoplasmic and nuclear Notch3 are increased by erlotinib, suggesting that at least a fraction of the Notch3 is activated. However, most of the beta catenin signal seems to remain in the cytoplasm. Is there an increase in nuclear beta catenin under erlotinib treatment, as transcript analysis suggests, and is this paralleled by an increase in nuclear Notch3? This is hard to quantify from the images presented, but it could be assessed by re-analyzing the images or, better, by directly measuring cytoplasmic and nuclear beta catenin and Notch3.

Notch3 causes β -catenin activation in a ligand-independent manner. Based on the observations presented in the manuscript, we show that Notch3 interacts with β -catenin predominantly in the cytoplasm, which leads to increased stability of β -catenin. We also performed cytoplasmic and nuclear fractionation experiments and found that EGFR TKI treatment led to increased nuclear β -catenin, both total and active forms and decreased phospho β -catenin (inactive) (**Fig. 4d**). Interestingly, we also found that EGFR TKI treatment led to decreased nuclear Notch3 and increased cytoplasmic Notch3, which is in agreement with our hypothesis and supports our notion that cytoplasmic association between Notch3 and β -catenin is helping β -catenin stability (**Fig. 4d**). Overall, these findings demonstrate the non-canonical role for cytoplasmic Notch3 where it activates β -catenin signaling. The decrease in Notch3 in the nucleus would argue against a role for canonical Notch3 signaling or for a role where Notch3 binds with β -catenin at responsive promoters.

- Minor points:

- o Line 100: there are numerous, well characterized non-canonical functions of Notch receptors in addition to interaction with beta catenin

Changed to: One of the non-canonical activities of the Notch1 receptor is its effect on β -catenin activity.

- o Line 234: Myc is also a Notch transcriptional target, not an exclusive beta catenin target.

Changed to: In addition, we see higher protein levels c-Myc and the stem cell markers Nanog and Oct4.

Response to reviewer 2:

1- The authors main novel findings in NSCLC are: (a) the interaction between β -catenin and Notch-3, and (b) the correlation between serum PAI1 level and EGFR TKI

resistance.

Regarding the former, literature data show an opposite correlation between b-catenin activity and Notch-3 interaction in other cancer types such that Notch-3 inhibits b-catenin function. As the authors stated, Notch-3 stabilizes beta-catenin in the cytoplasm and indeed co-IF stains show cytoplasmic beta-catenin and Notch-3 (Figs. 3 C-D and 7 A-B). How do the authors reconcile their data with the literature and how do they link this finding to b-catenin function since it is expected that b-catenin function (transcriptional activity) requires its nuclear instead of cytoplasmic localization.

Notch1 has been demonstrated to inhibit β -catenin activity in a non-canonical manner, and specifically Notch1 is well known to have opposite effects in different contexts. Notch3 is less understood and may have effects unique to the lung cancer context. There is a paper on HCC where the authors show an inverse correlation between Notch3 expression and β -catenin expression, but they do not show association by co-immunoprecipitation (Zhang, et al., *Oncotarget*, 2015 Feb; 6(6); 3669). Our co-immunoprecipitation data does show that Notch3 interacts with β -catenin in an EGFR TKI dependent manner, either directly or in a complex, which appears to be in the cytoplasm. We also show that this association increases with EGFR TKI treatment. Since cytoplasmic β -catenin is targeted for proteolysis by a group of proteins referred as β -catenin destruction complex, our data suggest the Notch3 association with β -catenin in the cytoplasm could be protecting β -catenin from the degradation thus leading to increased stability. To further demonstrate mechanical insights on how EGFR TKI treatment leads to β -catenin activation, we performed a cytoplasmic and nuclear fractionation experiment. Fractionation analysis clearly demonstrated that EGFR TKI treatment causes nuclear accumulation of the active form of β -catenin while also decreasing phospho β -catenin (inactive form which is the target for destruction) (**Fig. 4d**).

Regarding the latter, high PAI1 expression was detected in one cell line (HCC4006) post TKI. Do other cell lines increase PAI1 post-TKI in a beta-catenin dependent fashion? In fact serum PAI1 level is already high in a significant patient cohort pre-TKI (Fig. 10). These data do not support a specific role for rapid, TKI-induced increases in PAI1 (via b-catenin activation) in patients. Additionally, what other factors beyond b-catenin regulate PAI1? How specific is PAI1 expression as a biomarker of b-catenin activation during EGFR TKI treatment?

Additional experiments were conducted in a total of 4 cell lines and demonstrated that EGFR TKI treatment induces soluble PAI1 in the cell culture medium (**Supplementary Fig. 6**). To further demonstrate that EGFR TKI mediated PAI1 is β -catenin dependent new data was added showing that EGFR TKI mediated PAI1 was sensitive to the β -catenin inhibitor, ICG001. We agree with the reviewer that some patients do have high levels of basal PAI1, which could be due to high baseline β -catenin signaling in EGFR mutant NSCLC. However, we do show that those patients with a large induction of PAI1 (i.e., low prior to treatment, and high post treatment,

Figs 10C and D) do worse than patients with low or no induction (Fig 10E). Based on the literature TGF-B1 can also induce PAI1 expression. The specificity of the PAI1 as a surrogate marker for the EGFR TKI treatment is predominantly through β -catenin activation. Our *in vitro* data showing that a β -catenin inhibitor reduces EGFR TKI induced PAI1 supports this (Fig 4A). However, we believe that tumors that escape EGFR TKI treatment in a Notch3 and β -catenin signaling independent manner may not show PAI1 expression. There is also possibility that other β -catenin targets such as MMP7 may serve as surrogate biomarkers.

2- The authors state that activating mutations in the Wnt/b-catenin pathway are rare in NSCLC. Recent published data indicate that this may not be the case, particularly in the setting of EGFR TKI disease persistence and resistance (PMIDs: 28854272, 29106415). In this light, the authors main finding, while interesting, is not especially novel.

We apologize for unintentionally missing these two relevant articles to cite in our manuscript. We would like to emphasize that our studies are in agreement with these investigations and complement the literature on β -catenin and EGFR TKI resistance. Our studies focus on short term EGFR TKI induced β -catenin activation without mutation. The aim of our study was to understand the early events of EGFR TKI treatment that leads to the development of EGFR TKI resistance in a β -catenin dependent manner. The mechanism we describe is completely novel where an association of Notch3 with b-catenin leads to β -catenin stabilization and allows for survival of a population of cells early in the treatment.

3- A central conclusion of this study is that combined treatment of EGFR TKI + b-catenin inhibitor is more effective than TKI monotherapy. Other papers have demonstrated the efficacy of this approach, as the authors mentioned. The novelty here is limited. Additionally, the effects of the b-catenin inhibitor alone are not shown *in vivo* nor is the molecular validation in tumors of its on-target effects and whether the effects can be rescued through constitutive activation of b-catenin (validating the on-target activity of the drug in these systems).

As per reviewers request, data for β -catenin inhibitor alone is included (**Fig. 9a & b**). Previous studies have used EGFR TKI in combination with ICG001 to target EGFR TKI acquired resistant models (EGFR T790M). However, we are the first to conduct preclinical studies using this combination in parental EGFR mutant NSCLC to reduce the development of resistance, and where it can impact a broader patient population. Again, we would like to emphasize that our focus is to target drug persisters that could lead to the development of resistance. On the other hand, the aim of the previous study was to target the resistant disease, which developed potentially due to the activation of drug persisters as we demonstrate. Overall, our study implicates that early treatment of EGFR mutant patients with the combination prevents the recurrence/resistance.

To demonstrate the specificity of β -catenin inhibitor, ICG-001, we have validated β -catenin transcriptional targets that were identified in this study using tumor samples from the pre-clinical studies presented in this manuscript. The gene expression data using qPCR analysis demonstrated that EGFR TKI treatment increases β -catenin transcriptional targets and further demonstrated that these β -catenin transcriptional targets were sensitive to the β -catenin inhibitor, ICG001 (**Supplementary Fig. 4**). This suggests that ICG-001 is able to inhibit the β -catenin signaling in our pre-clinical studies.

4- In general, the *in vivo* efficacy data show modest effect sizes. The magnitude of benefit of the combination therapy appears unlikely to correlate with significant improvements in the control of NSCLC in the clinic.

The overall survival (OS) and tumor recurrence analysis data are very strong. Also, we would like to point out that the overall survival and recurrence free survival (RFS) data were obtained after stopping the treatments. We believe that given prolonged treatments with optimized treatment conditions, the combination therapy may produce durable OS and RFS in patients with NSCLC.

It is also unclear how the *in vivo* data specifically demonstrate that it is induction of β -catenin versus selection for pre-existing cells with high β -catenin signaling during EGFR TKI treatment that leads to tumor progression over time.

We have clearly shown in this manuscript that treatment with an EGFR TKI induces β -catenin activity in a subset of cells in NSCLC cell lines, which is Notch3 dependent. In a sense, we are selecting for cells with high β -catenin signaling, but we can't be sure if there is some pre-existing population that has high baseline β -catenin activity, induced β -catenin activity, or both. One piece of evidence from our previous work that shows an increase in the ALDH positive cell population with EGFR TKI treatment, which was also Notch3 dependent. In this manuscript, ALDH1A staining of the tumors shows that there is a high level of ALDH1A protein (Fig 6A, B). While this is not absolute proof that the tumors are arising from our induced population, it suggests that this is the case.

5- β -catenin and EGFR TKI resistance have been linked to EMT. Have the authors examined this connection in their systems? Do the patient specimens show evidence of an EMT, particularly in association with EGFR TKI resistance and β -catenin activation.

We have performed IHC analysis of EMT markers *in vivo* using xenograft tumor samples that were treated with control or EGFR TKI and demonstrated that EGFR TKI leads to upregulation of EMT markers (**Fig. 6c, d & e**). Our data have uncovered a specific mechanism for β -catenin induction that is targetable with drugs in the clinic, unlike EMT.

Reviewers' comments:

Reviewer #1 (Remarks to the Author):

The revised manuscript addresses all my questions, and it presents a consistently convincing case for a novel non-canonical function of Notch3 in NSCLC. These findings have novel mechanistic implications and significant translational relevance.

Minor issues:

The lane labels in Figure 3c are misaligned

The "DMSO" caption in Figure 3d is misaligned

Reviewer #2 (Remarks to the Author):

The authors have submitted a revised manuscript that is improved overall. However, several concerns remain unaddressed or addressed with insufficient mechanistic depth to reach firm conclusions.

(1) The authors fail to provide clarity on the precise mechanistic basis of how Notch3 interaction can simultaneously bind to b-catenin in the cytoplasm yet also promote increased nuclear accumulation of b-catenin. This is an unresolved paradox that call into question the scientific premise of the main conclusion of the study.

(2) Regarding novelty of the observation of enhanced anti-cancer activity of co-suppression of b-catenin signaling along with EGFR TKI treatment: the authors seem to be unaware of prior published works such as PMID: 22738915. This report showed the benefit of combined pathway inhibition in the initial treatment context including in EGFR inhibitor sensitive lung cancer cell lines, and not only in cells with acquired EGFR inhibitor resistance. Thus, the translational hypothesis for this combination therapy in the current manuscript is not particularly innovative.

(3) The authors failed to address point #5 regarding the clinical specimen analysis. This remains an important open question that is germane to the potential clinical relevance of the current findings.

Response to Reviewers' comments:

We appreciate your positive comments on our manuscript. We would like to thank the reviewers again for their comments regarding our manuscript titled **“Notch3-dependent β -catenin signaling mediates EGFR TKI drug persistence in EGFR mutant NSCLC”**. We have addressed the concerns raised by the reviewer 2 in a detailed point-by-point response below. We strongly feel that you will find the revised manuscript satisfactory for publication in Nature Communications.

Reviewers' comments:

Reviewer #1 (Remarks to the Author):

The revised manuscript addresses all my questions, and it presents a consistently convincing case for a novel non-canonical function of Notch3 in NSCLC. These findings have novel mechanistic implications and significant translational relevance.

We would like to thank reviewer 1 for his/her constructive criticism, which helped improve the quality of our work.

Minor issues:

The lane labels in Figure 3c are misaligned

The "DMSO" caption in Figure 3d is misaligned

We appreciate that reviewer 1 identified problems in the alignment of Figure 3c and 3d, which have been fixed.

Reviewer #2 (Remarks to the Author):

The authors have submitted a revised manuscript that is improved overall. However, several concerns remain unaddressed or addressed with insufficient mechanistic depth to reach firm conclusions.

(1) The authors fail to provide clarity on the precise mechanistic basis of how Notch3 interaction can simultaneously bind to β -catenin in the cytoplasm yet also promote increased nuclear accumulation of β -catenin. This is an unresolved paradox that call into question the scientific premise of the main conclusion of the study.

We do not suggest that Notch3 interaction can simultaneously bind to β -catenin in the cytoplasm yet also promote increased nuclear accumulation of β -catenin. We rather show an increased stability of activated β -catenin due to Notch3 association that is inhibited by constitutively activated EGFR and released upon treatment with an EGFR TKI, resulting in an EGFR TKI dependent role for Notch3 in the activation of β -catenin transcriptional activity in these cells. Mechanistically, we have clearly demonstrated that Notch3 associates with β -catenin in the

cytoplasm and increases the stability of the activated β -catenin. Increased levels of activated β -catenin then increase total β -catenin localized to the nucleus where it activates transcription. We do not have data nor propose that Notch3 promotes β -catenin translocation.

(2) Regarding novelty of the observation of enhanced anti-cancer activity of co-suppression of β -catenin signaling along with EGFR TKI treatment: the authors seem to be unaware of prior published works such as PMID: 22738915. This report showed the benefit of combined pathway inhibition in the initial treatment context including in EGFR inhibitor sensitive lung cancer cell lines, and not only in cells with acquired EGFR inhibitor resistance. Thus, the translational hypothesis for this combination therapy in the current manuscript is not particularly innovative.

We apologize the reviewer 2 for not mentioning the article PMID: 22738915, which demonstrated the role of canonical Wnt signaling for cell survival during EGFR TKI treatment. We agree with the reviewer that this is not the first study to hypothesize the clinical utility of targeting the EGFR and Wnt/ β -catenin pathways to manage EGFR mutant NSCLC. However, our study is entirely different from the article PMID: 22738915 or from the other published articles related to drug resistance to EGFR TKI and β -catenin and is innovative for the following reasons:

- This study is innovative because for the first time we illustrated the role of β -catenin and its novel regulation through Notch3 in EGFR TKI induced drug persister cells using *in vitro* and preclinical models.
- This study identifies a specific population of cells which we call “drug persister cells” which have high tumorigenic potential and have the ability to survive during EGFR TKI therapy, thus enabling the development of EGFR TKI mediated drug resistance.
- We further established a clinical relevance of β -catenin activation to EGFR TKI resistance using patient tumor samples.
- Identified PAI1 as a non-invasive, mechanism-based biomarker that further connects with β -catenin activation especially in EGFR TKI resistant NSCLC patients.
- We have validated the combination of an EGFR TKI and a β -catenin inhibitor in preclinical models which demonstrated the benefit in recurrence free survival and overall survival.
- Our study also demonstrates that the drug persister cells have a “cancer stem cell like phenotype” which explains the potential role of stemness in the development of drug resistance to EGFR TKI therapy.

(3) The authors failed to address point #5 regarding the clinical specimen analysis. This remains an important open question that is germane to the potential clinical relevance of the current findings.

The EMT phenotype is a pattern of cellular biological features and gene expression alterations that may be found in tumors with EGFR resistance, but does not bear on the specific mechanistic pathways that we have identified in this manuscript. Our study focuses on changes that happen within days of initiation of EGFR TKIs, and the available human samples are from biopsies taken at the time of clinical resistance, months or years after drug initiation. Thus, our study does not focus on, and is not dependent on, demonstration of the EMT phenotype, about which there are

already multiple correlational studies in tumors at the time of clinical resistance. Thus, we do not believe that determining the correlation of Notch3- β -catenin functions with the EMT phenotype observable at in human tumors at the time of clinical resistance months or years after therapy initiation is relevant to the point of this manuscript.